# Thresholds for sensitive optimality and Blackwell optimality in stochastic games

**Stéphane Gaubert**
INRIA and Ecole Polytechnique
stephane.gaubert@inria.fr

**Julien Grand-Clément**
HEC Paris
grand-clement@hec.fr

**Ricardo D. Katz**
CIFASIS-CONICET
katz@cifasis-conicet.gov.ar

## Abstract

We investigate refinements of the mean-payoff criterion in two-player zero-sum perfect-information stochastic games. A strategy is *Blackwell optimal* if it is optimal in the discounted game for all discount factors sufficiently close to $1$. The notion of *$d$-sensitive optimality* interpolates between mean-payoff optimality (corresponding to the case $d = -1$) and Blackwell optimality ($d = \infty$). The *Blackwell threshold* $\alpha_{\mathsf{Bw}} \in [0, 1[$ is the discount factor above which all optimal strategies in the discounted game are guaranteed to be Blackwell optimal. The *$d$-sensitive threshold* $\alpha_{\mathsf{d}} \in [0, 1[$ is defined analogously. Bounding $\alpha_{\mathsf{Bw}}$ and $\alpha_{\mathsf{d}}$ are fundamental problems in algorithmic game theory, since these thresholds control the complexity for computing Blackwell and $d$-sensitive optimal strategies, by reduction to discounted games which can be solved in $O\left((1 - \alpha)^{-1}\right)$ iterations. We provide the first bounds on the $d$-sensitive threshold $\alpha_{\mathsf{d}}$ beyond the case $d = -1$, and we establish improved bounds for the Blackwell threshold $\alpha_{\mathsf{Bw}}$. This is achieved by leveraging separation bounds on algebraic numbers, relying on Lagrange bounds and more advanced techniques based on Mahler measures and multiplicity theorems.

## 1   Introduction

Two-player perfect-information zero-sum stochastic games (SGs) are an important class of Shapley's stochastic games [Sha53] where every state is controlled by a unique player. SGs are a cornerstone of game theory, with important applications in auctions [LLP$^+$99], mechanism design [Nar14], multi-agent reinforcement learning [Lit94], robust optimization [GCPV23, CGK$^+$23], and $\mu$-calculus model-checking [CHV$^+$18]. Shapley originally considered the sum of discounted instantaneous rewards as the objective; the mean-payoff objective was studied in [Gil57]. Stationary and deterministic optimal strategies exist for these objectives in the perfect-information case [Sha53, LL69].

The mean-payoff objective coincides with the limit of the discounted objective as the discount factor goes to $1$. This suggests to study the variation of the set of optimal strategies in the discounted game, as a function of the discount factor. Blackwell showed (in the one-player case) that, for finite state and action models, there exist strategies that remain discount optimal for all values of the discount factor that are sufficiently close to $1$ [Bla62]. These are now called *Blackwell optimal strategies*. As the discount factor approaches $1$, the "weight" given to the rewards received in the distant future increases. Therefore, Blackwell optimal strategies can be considered as the most farsighted (or least greedy) strategies for models in long horizon. This is why better understanding Blackwell optimality

39th Conference on Neural Information Processing Systems (NeurIPS 2025).

is referred to as "*one of the pressing questions in reinforcement learning*" in [DDE+20]. Veinott introduced (still in the one-player case) the notion of $d$-*sensitive optimality*, interpolating between mean-payoff optimality (obtained for $d = -1$) and Blackwell optimality (obtained for $d = \infty$). The value associated with a given stationary strategy has a Laurent series expansion in the powers of $1 - \alpha$, where $\alpha$ is the discount factor, with a pole of order at most $-1$ at $\alpha = 1$. Veinott's $d$-sensitive objective involves the sequence of coefficients of this expansion, up to order $d$, which is maximized or minimized with respect to the lexicographic order (see Theorem 10.1.6 in [Put14]). In particular, any $d$-sensitive optimal strategy is $d'$-sensitive optimal for all $d' < d$. Moreover, for models with $n$ states, every $(n - 2)$-sensitive optimal strategy is Blackwell optimal. The notion for $d = 0$ is also known as *bias optimality*.

It is worth emphasizing that the notions of Blackwell optimality and mean-payoff optimality have received increased attention recently in the reinforcement learning community, see [DDE+20, DG22, TRMV21, YGA+16]. More generally, computing mean-payoff optimal strategies is an important open question in algorithmic game theory and has been extensively studied. *Pseudo*-polynomial algorithms exist for mean-payoff *deterministic* instances, based on *pumping* [GKK88] and value iteration [ZP96, CGB03, GS08]. In more generality, computing mean-payoff or bias optimal strategies is difficult, since the Bellman operator is no longer a contraction when $\alpha = 1$, and the mean-payoff and bias objectives may be discontinuous in the entries of stationary strategies (e.g., Chapter 4 of [FS12]). In fact, strategy iteration may cycle for the mean-payoff objective in multichain instances (see Section 6 of [ACTDG12]), and other algorithms have an exponential dependence on the number of states with stochastic (non-deterministic) transitions or undetermined complexities [ACTDG12, BEGM10]. Besides the method based on the Blackwell threshold discussed below, the only algorithm we are aware of to compute Blackwell optimal strategies consists in performing policy iteration, considering the discount factor as a formal parameter, and encoding the discounted value function associated with a pair of strategies by its Laurent series expansion truncated at order $n - 2$, leading to a nested lexicographic policy iteration method, see [Put14, Chapter 10.3] for a presentation in the one player case. A counter example of Friedmann implies that for deterministic games, this algorithm can take an exponential time [Fri09].

The existence of the *Blackwell threshold*, defined as the smallest number $\alpha_{\mathsf{Bw}} \in [0, 1[$ such that strategies that are optimal for any discount factor $\alpha \geq \alpha_{\mathsf{Bw}}$ are Blackwell optimal, is well-known, e.g. [AM09]. We define the $d$-*sensitive threshold* as the smallest number $\alpha_{\mathsf{d}} \in [0, 1[$ such that strategies that are discount optimal for any discount factor $\alpha \geq \alpha_{\mathsf{d}}$ are $d$-sensitive optimal. The existence of $\alpha_{\mathsf{Bw}}$ and $\alpha_{\mathsf{d}}$ suggests a simple method for computing Blackwell and $d$-sensitive optimal strategies: compute discount optimal strategies for a discount factor $\alpha$ close to 1. This approach has the advantage that *any* advances in solving discounted SGs transfer to algorithms for Blackwell and mean-payoff optimal strategies. Note that the term $(1 - \alpha)^{-1}$ controls the complexity of computing discount optimal strategies: for known game parameters (rewards and transitions), strategy iteration and value iteration scale as $\tilde{O}((1 - \alpha)^{-1})$ [HMZ13] (hiding the dependence on the number of states/actions, and lower order terms in $1 - \alpha$). For unknown game parameters, model-based methods [ZKBY23] or model-free methods based on Q-learning [SWYY20] scale as $\tilde{O}((1 - \alpha)^{-3})$.

Thus, for computational purposes, it is crucial to upper bound $\alpha_{\mathsf{Bw}}$ and $\alpha_{\mathsf{d}}$, or equivalently to lower bound $1 - \alpha_{\mathsf{Bw}}$ and $1 - \alpha_{\mathsf{d}}$. For such bounds to be useful, they should only rely on a few game parameters that are available by design. In this paper, we consider SGs that satisfy the following assumption.

**Assumption 1.1.** $\Gamma$ *is a perfect-information stochastic game with finitely many states and actions, integer rewards, and rational transition probabilities. The parameters of the game $\Gamma$ are $n$, $W$ and $M$, and are defined as follow:*

- $n \in \mathbb{N}$ *is the number of states,*

- $W \in \mathbb{N}$ *is an upper bound on the absolute values of the integer rewards,*

- $M \in \mathbb{N}$ *is the common denominator of the transition probabilities.*

**Main results.** We provide upper bounds for the Blackwell threshold $\alpha_{\mathsf{Bw}}$ and for the $d$-sensitive threshold $\alpha_{\mathsf{d}}$ for perfect-information SGs. We also derive stronger results in the case of deterministic games (transition probabilities in $\{0, 1\}$) or for stochastic games with unichain structure.

At the core of our results are separation bounds for algebraic numbers, a classical topic in algebraic number theory. More precisely, we use three separation methods. The first method relies on a result of Lagrange [Lag69], also obtained by Hadamard [Had93] and Fujiwara [Fuj16], providing a lower bound on the modulus $|z|$ of any non-zero root $z$ of a polynomial $P$, see Lecture IV in [Yap00]. The second method relies on *Mahler measures* to lower bound $|1 - z|$ when $z \neq 1$ is a root of a polynomial $P$, especially on a theorem of Dubickas [Dub95] building on a series of earlier works based on the seminal paper by Mignotte and Waldschmidt [MW94]. The third method uses a bound of Borwein, Edérlyi, and Kós on the multiplicity of 1 as a root of a polynomial with integer coefficients [BEK99], which controls the value of $d$ such that $\alpha_d = \alpha_{Bw}$. We shall see that each of these approaches yields a useful bound – not dominated by the other bounds in some regimes of the game parameters $n$, $W$ and $M$. We present our main results below, according to the different techniques. For the sake of readability, we simplify some of our results with $O(.)$ notations here, and we defer the detailed statements to the next sections. We start with the results based on Lagrange.

**Theorem 1.2** (Based on Lagrange bound). *If the game $\Gamma$ satisfies Assumption 1.1, then:*

Deterministic case ($M = 1$). *The d-sensitive threshold $\alpha_d$ and the Blackwell threshold $\alpha_{Bw}$ satisfy*

$$\alpha_d \leq 1 - \frac{1}{24W\binom{2n}{\min\{d+4,n\}}} \quad and \quad \alpha_{Bw} \leq 1 - \frac{1}{24W\binom{2n}{n}} . \tag{1}$$

Stochastic case ($M > 1$). *The Blackwell threshold $\alpha_{Bw}$ satisfies* $\alpha_{Bw} \leq 1 - \dfrac{2^{\lfloor \frac{2}{3}n \rfloor - 2}}{nW(2M)^{2n-1}\binom{2n-1}{\lfloor \frac{2}{3}n \rfloor}}$.

*If $\Gamma$ is unichain, the d-sensitive threshold $\alpha_d$ satisfies* $\alpha_d \leq 1 - \dfrac{2^{\min\{d+2,\lfloor \frac{2}{3}n-1 \rfloor\}-1}}{nW(2M)^{2n-1}\binom{2n-1}{\min\{d+2,\lfloor \frac{2}{3}n-1 \rfloor\}+1}}$.

Our next set of results are bounds on $\alpha_{Bw}$ based on Mahler measures. Since these bounds are more difficult to read than the ones in the previous theorem, we only give the resulting $O(\cdot)$ expressions here, and we provide the exact values in Sections 3 and 4. We provide bounds on $-\log(1 - \alpha_{Bw})$, i.e., on the value of $L > 0$ such that $\alpha_{Bw} \leq 1 - e^{-L}$.

**Theorem 1.3** (Based on Mahler measures). *If the game $\Gamma$ satisfies Assumption 1.1, then:*

Deterministic case. *The Blackwell threshold $\alpha_{Bw}$ satisfies*

$$-\log(1 - \alpha_{Bw}) \leq O\left(\max\left\{\sqrt{n\log(n)\log(\sqrt{n}W)}, \log(\sqrt{n}W)\right\}\right) .$$

Stochastic case. *The Blackwell threshold $\alpha_{Bw}$ satisfies*

$$-\log(1 - \alpha_{Bw}) \leq O\left(\max\left\{\log(W) + n(1 + \log(M)), \sqrt{n\log(n)(\log(W) + n(1 + \log(M)))}\right\}\right).$$

We now present our last set of results. [Vei69] shows that, for Markov decision processes (MDPs, i.e., the one-player case), $(n - 2)$-sensitive optimal strategies are also Blackwell optimal. Our next theorem shows that a much smaller value of $d$ may suffice.

**Theorem 1.4.** *Assume the game $\Gamma$ satisfies Assumption 1.1 and is deterministic. Then every $\bar{d}^{det}(n, W)$-sensitive optimal strategy is Blackwell optimal, where $\bar{d}^{det}(n, W) = O\left(\sqrt{n\log(W)}\right)$.*

When $\log(W) = o(n)$, $\bar{d}^{det}(n, W)$ may be much smaller than $n - 2$. Combining Theorem 1.4 with the bounds on $\alpha_d$ of Theorem 1.2, we arrive at the following bound on $\alpha_{Bw}$.

**Theorem 1.5** (Based on multiplicity). *Assume that $\Gamma$ satisfies Assumption 1.1 and is deterministic. Then the Blackwell threshold $\alpha_{Bw}$ satisfies* $\alpha_{Bw} \leq 1 - \dfrac{1}{24W\binom{2n}{\min\{O\left(\sqrt{n\log(W)}\right),n\}}}$ .

In the case of general (non-deterministic) perfect-information SGs, the multiplicity method of Theorem 1.5 does not lead to a useful bound. We further elaborate on this in the last part of Section 4. To simplify the comparison between our bounds, and with existing work, we summarize all our results in Tables 1 and 2. In these tables, we reformulate our bounds on $\alpha_{Bw}$ as bounds on $-\log(1 - \alpha_{Bw})$, i.e., on the value of $L > 0$ such that $\alpha_{Bw} \leq 1 - e^{-L}$, and similarly for $\alpha_d$.

Note that Assumption 1.1 is necessary to obtain a meaningful bound on $\alpha_{\mathsf{Bw}}$ and $\alpha_{\mathsf{d}}$ (Proposition 4.3 in [GCP23]). We emphasize that the bounds derived in Theorems 1.2, 1.3, and 1.5 are complementary, i.e., there are different regimes of $\log(W)$, $\log(M)$ and $n$ where one bound is better than the others. As an example, consider deterministic games ($M = 1$). When $W = O(1)$, Theorem 1.2 yields $-\log(1 - \alpha_{\mathsf{Bw}}) \leq O(n)$, while Theorems 1.3 and 1.5 both lead to the stronger bound $-\log(1 - \alpha_{\mathsf{Bw}}) \leq O(\log(n)\sqrt{n})$. In contrast, in the regime where $W = \exp(\Theta(n))$, Theorems 1.2 and 1.5 lead to the bound $-\log(1 - \alpha_{\mathsf{Bw}}) \leq O(n)$, which is stronger than Theorem 1.3 yielding $-\log(1 - \alpha_{\mathsf{Bw}}) \leq O\left(n\sqrt{\log(n)}\right)$. We provide more discussion on this in Sections 3 and 4.

Compared to previous work, we note that our bound on $-\log(1 - \alpha_{\mathsf{Bw}})$ from Theorem 1.2 improves upon [AM09] and [GCP23] by a factor $\Omega(n)$, while our bound from Theorem 1.5 recovers for SGs the results obtained in [MK25] for MDPs. Theorem 1.3 provides a new approach to bounding $\alpha_{\mathsf{Bw}}$ based on Mahler measures. We are also the first to provide bounds for $\alpha_{\mathsf{d}}$, beyond the case $d = -1$ for deterministic games analyzed in [ZP96]. However, we note that our results on $\alpha_{\mathsf{Bw}}$ do not lead to weakly- or even quasi-polynomial time algorithms for computing Blackwell optimal strategies: our bounds on $\alpha_{\mathsf{Bw}}$ are still very close to 1, and the algorithms with the best dependence on $\alpha$ for solving discounted SGs require $\tilde{O}((1 - \alpha)^{-1})$ arithmetic operations [HMZ13]. We refer to the next section for a detailed discussion and comparison with the existing bounds on $\alpha_{\mathsf{Bw}}$ and $\alpha_{\mathsf{d}}$.

Table 1: Bounds for *deterministic* SGs satisfying Assumption 1.1. For the sake of readability, in all these bounds we omit constant terms and the $O(\cdot)$ notation.

| | Bound on $-\log(1 - \alpha_{\mathsf{Bw}})$ | Bound on $-\log(1 - \alpha_{\mathsf{d}})$ | Remarks |
|---|---|---|---|
| [ZP96] | $\times$ | $\log(W) + 3 \cdot \log(n)$ | For $d = -1$ |
| [AM09] | $n\log(n) + n^2\log(W)$ | $\times$ | . |
| [GCP23] | $n^2 + n\log(W)$ | $\times$ | For MDPs |
| [MK25] | $\sqrt{n\log(W)}\log\left(\frac{n}{\log(W)}\right) + \log(W)$ | $\times$ | For MDPs |
| Th. 1.2 | $n + \log(\frac{W}{\sqrt{n}})$ | $\log(W) + (d+4)\left(1 + \log\left(\frac{2n}{d+4}\right)\right)$ | Lagrange |
| Th. 1.3 | $\max\{\sqrt{n\log(n)}\log(\sqrt{n}W), \log(\sqrt{n}W)\}$ | $\times$ | Mahler |
| Th. 1.5 | $\sqrt{n\log(W)}\log\left(\frac{n}{\log(W)}\right) + \log(W)$ | $\times$ | Multiplicity |

Table 2: Bounds for SGs satisfying Assumption 1.1. For the sake of readability, in all these bounds we hide the constant term and the $O(\cdot)$ notation. $^{\dagger}$: This bound holds only in the unichain case.

| | Bound on $-\log(1 - \alpha_{\mathsf{Bw}})$ | Bound on $-\log(1 - \alpha_{\mathsf{d}})$ | Remarks |
|---|---|---|---|
| [AM09] | $n\log(n) + n^2\log(\max\{W, M\})$ | $\times$ | . |
| [GCP23] | $n^2(1 + \log(M)) + n\log(W)$ | $\times$ | For MDPs |
| Th. 1.2 | $n(1 + \log(M)) + \log(W)$ | $\log(nW(2M)^{2n-1}) + (d+4)\log\left(\frac{2n}{d+4}\right)^{\dagger}$ | Lagrange |
| Th. 1.3 | $\max\{\sqrt{n\log(n)}F, F\}$ $F := n(1 + \log(M)) + \log(W)$ | $\times$ | Mahler |

**Related work.** To the best of our knowledge, only a small number of papers have obtained bounds on the $d$-sensitive threshold $\alpha_{\mathsf{d}}$ (only for $d = -1$) or on the Blackwell threshold $\alpha_{\mathsf{Bw}}$.

*d-sensitive threshold.* The existence of $\alpha_{-1}$ is proved in the seminal paper [LL69] and in Theorem 5.4.4 of [Pur95]. It is used, for instance, in [AM09] to show that these games polynomially reduce to solving discounted SGs. [ZP96] shows the tight bound $\alpha_{-1} \geq 1 - \frac{1}{8Wn^3}$ for deterministic perfect-information mean-payoff SGs. Theorem 1.2 recovers this bound for $d = -1$, but no general bound was known for $\alpha_{\mathsf{d}}$ (before our work). [AG13] shows that if there is a renewal state, i.e., a state for which the time of first return from any other state is bounded by $N \in \mathbb{N}$, then we can take $\alpha_{-1} = 1 - 1/N$. A characterization of $\alpha_{-1}$ for the one-player case (i.e., for MDPs) is given in [Boo23] but the obtained bound relies on the minimum gain difference between two strategies, instead of only the game parameters $n$, $M$ and $W$ as in our results. Finally, we note that a line of work studies

a related but different question for MDPs and reinforcement learning, namely, how close to 1 should $\alpha$ be for a discount optimal strategy to be $\epsilon$-optimal for the mean-payoff criterion, e.g. [WWY22] showing that $\alpha \geq 1 - \epsilon/H$ suffices for weakly-communicating instances with diameter $H \in \mathbb{N}$, or [JS21] using assumptions on the mixing times of the Markov chains induced by deterministic strategies. In contrast to these works, we focus on optimality (instead of $\epsilon$-optimality) and most of our results do not require an assumption on the chain structure (except for $\alpha_d$ in the stochastic case).

*Blackwell threshold.* The existence of $\alpha_{Bw}$ comes from an argument developed initially by Blackwell [Bla62]. The works closest to ours are [AM09, GCP23, MK25]. [AM09] shows a bound on $\alpha_{-1}$ for SGs but their argument extends directly to $\alpha_{Bw}$. [GCP23] gives a bound on $\alpha_{Bw}$ for MDPs. [MK25] shows an analog of Theorem 1.4 for *deterministic* MDPs using Lagrange's bound. We summarize these bounds in Tables 1 and 2. We also note that the Blackwell threshold has found recent applications in the study of robust MDPs [GCPV23, WVA+24] and in the smoothed analysis of the complexity of strategy iteration [LS24].

*Main improvements compared to previous works.* Our work appears to be the first to provide a bound on the $d$-sensitive threshold $\alpha_d$ beyond the case $d = -1$ (corresponding to mean-payoff optimality) in the deterministic setting. Regarding $\alpha_{Bw}$, [MK25] only focuses on deterministic MDPs using the multiplicity approach, i.e., on the one-player deterministic case, whereas we focus on two-player SGs. This line of analysis based on multiplicity (Theorem 1.4) does not extend to the non-deterministic case (see the end of Section 4), where the corresponding bound on the value of $d$ such that $\alpha_d = \alpha_{Bw}$ is much larger than $n - 2$, the bound proved in the seminal paper [Vei69]. Compared to [GCP23] and [AM09], our bounds from Theorems 1.2 and 1.3 compare favorably: for instance, the bounds on $-\log(1 - \alpha_{Bw})$ in [GCP23] and [AM09] are worse by a factor of $\Omega(n)$ compared to our bounds based on Lagrange. This improvement comes from two main innovations in the analysis: better bounds on the magnitudes of the coefficients of the considered polynomials, and the use of much stronger root separation results - [AM09] uses a bound attributed to Cauchy (e.g. Equation (5) in Lecture IV of [Yap00]) which is weaker than the one we use due to Lagrange, while [GCP23] uses a bound from [Rum79], which applies to the distance between any two roots of a polynomial, whereas we only need to separate a root from 1 (and not from any other conjugates).

**Outline.** This paper is organized as follows. We introduce perfect-information SGs in Section 2. We derive bounds for the deterministic case in Section 3 and bounds for the stochastic case in Section 4. The implications of our results are discussed in Section 5. All the proofs can be found in the appendix.

# 2 Preliminaries on perfect-information stochastic games

Lloyd Shapley introduced SGs in the seminal paper [Sha53]. We focus on the case of two-player zero-sum SGs with finitely many states and actions, and discrete time. We denote the state space by $[n] := \{1, \ldots, n\}$ for some $n \in \mathbb{N}$. At every stage $k \in \mathbb{N}$, the game is in a state $i_k \in [n]$ observable by both players, called Min and Max. An instantaneous reward of $r_i^{ab}$ is determined when Min chooses action $a$ and Max chooses action $b$ in state $i$, and then the game transitions to the next state $j \in [n]$ with probability $P_{ij}^{ab} \in [0, 1]$. Thus, we consider SGs where the rewards $r_i^{ab}$ depend on the current state $i$ and the players' actions $a, b$, but not on the next state $j$. It is worth noting that this assumption is standard and can be made without loss of generality (any game with rewards depending on the next state $j$, i.e., of the form $r_{ij}^{ab}$, can be converted to a game with rewards of the form $r_i^{ab}$ expanding the state space to include intermediary states). We focus on the case of *perfect-information* SGs, where we can partition the state space $[n]$ into the states controlled by Min and those controlled by Max.

A *strategy* of a player is a function that assigns to a history of the game (i.e., the sequence of previous states and actions) a decision (choice of an action) of this player. A pair of strategies $(\sigma, \tau)$ of players Min and Max induces a probability measure on the set of sequences of states. We define the *discounted value function* $\alpha \mapsto v_i^{\sigma,\tau}(\alpha)$ from $]0, 1[$ to $\mathbb{R}$, associated with the pair of strategies $(\sigma, \tau)$ and the initial state $i$, as $v_i^{\sigma,\tau}(\alpha) := \mathbb{E}^{\sigma,\tau} \left[ \sum_{k=0}^{+\infty} \alpha^k r_{i_k}^{a_k b_k} \mid i_0 = i \right]$ where $r_{i_1}^{a_1 b_1}, r_{i_2}^{a_2 b_2}, \ldots$ is the random sequence of instantaneous rewards induced by $(\sigma, \tau)$.

**Definition 2.1.** A pair of strategies $(\sigma^*, \tau^*)$ of players Min and Max is *discount optimal* for the discount factor $\alpha$ if for any state $i$ and for any strategies $\sigma$ and $\tau$ of players Min and Max, we have

$$v_i^{\sigma, \tau^*}(\alpha) \geq v_i^{\sigma^*, \tau^*}(\alpha) \geq v_i^{\sigma^*, \tau}(\alpha) .$$

A pair of strategies $(\sigma^*, \tau^*)$ is *Blackwell optimal* if there exists $\bar{\alpha} < 1$ such that $(\sigma^*, \tau^*)$ is discount optimal for all discount factors larger than $\bar{\alpha}$.

Shapley [Sha53] shows the existence of stationary discount optimal strategies. If, in addition, the game is a perfect-information SG, then these stationary optimal strategies may be chosen deterministic. The existence of stationary deterministic Blackwell optimal strategies for perfect-information SGs is a consequence of the analysis in the seminal paper [LL69] (see the proof of Theorem 1 in [LL69]).

**Definition 2.2.** A pair of strategies $(\sigma^*, \tau^*)$ of players Min and Max is *d-sensitive optimal* if

$$\lim_{\alpha \to 1^-} (1-\alpha)^{-d} \left( v_i^{\sigma, \tau^*}(\alpha) - v_i^{\sigma^*, \tau^*}(\alpha) \right) \geq 0, \ \lim_{\alpha \to 1^-} (1-\alpha)^{-d} \left( v_i^{\sigma^*, \tau^*}(\alpha) - v_i^{\sigma^*, \tau}(\alpha) \right) \geq 0 \ (2)$$

for any state $i$ and any strategies $\sigma$ and $\tau$ of players Min and Max respectively.

Our definition of $d$-sensitive optimality for two-player SGs recovers the definition of $d$-sensitive optimality for the one-player case [Vei69]. Note that Blackwell optimal strategies are $d$-sensitive optimal for $d = -1, 0, \ldots$. In fact, the same analysis as for MDPs (using the Laurent series expansion, e.g. Theorem 10.1.6 of [Put14]) shows that $n-2$-sensitive optimal strategies are Blackwell optimal, and mean-payoff optimality corresponds to $d = -1$.

Before diving into our main results, we give an **overview of our main proof techniques**.

*Bounds on $\alpha_{\mathsf{Bw}}$.* Our bounds are based on studying the zeros of the functions $\alpha \mapsto v_i^{\sigma, \tau}(\alpha) - v_i^{\sigma', \tau'}(\alpha)$ for any pairs $(\sigma, \tau)$ and $(\sigma', \tau')$ of stationary strategies and any state $i$. This approach has been used for MDPs, e.g., see Theorem 2.16 of [FS12]. Differences of discounted value functions are rational functions in $\alpha \in ]0, 1[$, i.e., they can be written as the ratio of two polynomials in $\alpha$. As such, each of them can only have finitely many zeros in $]0, 1[$. Additionally, for perfect-information SGs, discount optimal strategies may be chosen stationary and *deterministic* [Gil57], hence we can consider only finitely many pairs of strategies. Therefore, there exists a discount factor $\bar{\alpha} < 1$ such that none of the functions $\alpha \mapsto v_i^{\sigma, \tau}(\alpha) - v_i^{\sigma', \tau'}(\alpha)$ has a zero in $]\bar{\alpha}, 1[$, and so *all* these functions have constant sign in $]\bar{\alpha}, 1[$. This guarantees that $\alpha_{\mathsf{Bw}} \leq \bar{\alpha}$, since strategies that are discount optimal for some $\alpha$ satisfying $\bar{\alpha} < \alpha < 1$ remain discount optimal for all larger discount factors (otherwise some function $\alpha \mapsto v_i^{\sigma, \tau}(\alpha) - v_i^{\sigma', \tau'}(\alpha)$ would change sign in $]\bar{\alpha}, 1[$). Thus, a way to bound $\alpha_{\mathsf{Bw}}$ is to determine how close to 1 can a zero of the functions $\alpha \mapsto v_i^{\sigma, \tau}(\alpha) - v_i^{\sigma', \tau'}(\alpha)$ be. Given Assumption 1.1, we can bound the degree of the numerator of $\alpha \mapsto v_i^{\sigma, \tau}(\alpha) - v_i^{\sigma', \tau'}(\alpha)$ and the size of its coefficients. We then use *root separation* results to separate the root of a polynomial from a given scalar. Compared to previous work, our first improvement lies in a better analysis of the coefficients of the considered polynomials. As our second and main improvement, we use stronger separation bounds than in previous work, a bound due to Lagrange (Theorem 3.2) and a bound based on Mahler measures (Theorem 3.7). [AM09] uses a weaker bound due to Cauchy, and [OB21] also uses the Cauchy bound for a different purpose (bounding the variations in value functions when $\alpha \to 1$).

*Deterministic vs. non-deterministic games.* For *deterministic* SGs, each transition probability belongs to $\{0, 1\}$. In this case, the discounted value function associated with a pair of stationary strategies and a state can be represented by the concatenation of a path and an elementary circuit in the graph of the game, see Section 3. This can be used to analyze the degree and the magnitude of the coefficients of the numerator of $\alpha \mapsto v_i^{\sigma, \tau}(\alpha) - v_i^{\sigma', \tau'}(\alpha)$. When the game is not deterministic, we rely on the closed-form expression of the discounted value functions using cofactor matrices, see Section 4.

*Bounds on the $d$-sensitive threshold $\alpha_{\mathsf{d}}$.* Our bounds on $\alpha_{\mathsf{d}}$ rely on the same proof techniques as for $\alpha_{\mathsf{Bw}}$, except that the $d$-sensitive optimality allows us to deduce more information on the coefficients of the considered polynomials (more precisely, that some of them are zero). Again, the deterministic case is easier because we can exploit the graph representation of the discounted value functions mentioned above, whereas for the case of non-deterministic SGs we require a unichain assumption.

# 3 Results for perfect-information deterministic stochastic games

We start by analyzing the structure of the discounted value functions in deterministic perfect information SGs. A deterministic game can be represented by a weighted digraph: its set of nodes is the set of states, there is an edge from state $i$ to state $j$ if some actions $a$ and $b$ of the players realize this transition, in which case this edge has weight $r_i^{ab}$. Given an initial state $i$, a pair of stationary strategies $(\sigma, \tau)$ determines a run of the game of the form "$\pi\gamma$", in which $\pi$ is a path and $\gamma$ is a circuit. Given a path $\pi = (i_0, \ldots, i_k)$, we set $\langle r, \pi\rangle_\alpha := r_{i_0 i_1} + \alpha r_{i_1 i_2} + \cdots + \alpha^{k-1} r_{i_{k-1} i_k}$ (for simplicity, we denote instantaneous rewards by $r_{ij}$ instead of $r_i^{ab}$). With this notation, $v_i^{\sigma,\tau}(\alpha) = \langle r, \pi\rangle_\alpha + \frac{\alpha^p}{1-\alpha^q}\langle r, \gamma\rangle_\alpha$, where $p$ is the length of $\pi$ (i.e., the number of edges it contains) and $q$ is the length of $\gamma$. To study the zeros of $\alpha \mapsto v_i^{\sigma,\tau}(\alpha) - v_i^{\sigma',\tau'}(\alpha)$, we analyze the roots of the polynomial

$$\Delta(\alpha) := (1 - \alpha^q)(1 - \alpha^{q'})(v_i^{\sigma,\tau}(\alpha) - v_i^{\sigma',\tau'}(\alpha)) \tag{3}$$

(here $q'$ is the length of the circuit associated with $(\sigma', \tau')$). The next lemma bounds the degree and coefficients of $\Delta$.

**Lemma 3.1.** *We can write $\Delta(\alpha) = \sum_{k=0}^K a_k \alpha^k$, where $|a_k| \leq 12W$ and $K \leq 2n - 1$.*

As described in the previous section, our bounds on $\alpha_{\mathsf{Bw}}$ and $\alpha_{\mathsf{d}}$ rely on separating the roots of the polynomial $\Delta$ from 1. We proceed to do so in the next section.

## 3.1 Separation based on the Lagrange bound

In this section, we rely on the *Lagrange bound* (e.g. Lemma 5 in Lecture IV of [Yap00]).

**Theorem 3.2** (Lagrange bound). *Let $P = \sum_{k=j}^d c_k x^k$ with $c_j \neq 0$. Then, any non-zero root $z$ of $P$ satisfies $|z| \geq \frac{1}{2} \min_{i \in \{j+1, \ldots, d\}, c_i \neq 0} (|c_j|/|c_i|)^{\frac{1}{i-j}}$.*

The Lagrange bound separates the non-zero roots of a polynomial $P$ from 0. To separate the roots of $\Delta$ from 1, we apply this bound to the polynomial $\epsilon \mapsto \Delta(1 - \epsilon)$. Using this approach, we prove the first part of Theorem 1.2, namely, the bound on $\alpha_{\mathsf{d}}$ for deterministic perfect-information SGs.

**Theorem 3.3.** *Assume the game $\Gamma$ satisfies Assumption 1.1 and is deterministic ($M = 1$). Then, the $d$-sensitive threshold $\alpha_{\mathsf{d}}$ satisfies $\alpha_{\mathsf{d}} \leq 1 - \dfrac{1}{24W\binom{2n}{\min\{d+4, n\}}}$ .*

The binomial coefficient in our bound on $\alpha_{\mathsf{d}}$ appears because of the change of variable $\epsilon = 1 - \alpha$ in the polynomial $\Delta$, necessary to apply the Lagrange bound to the polynomial $\epsilon \mapsto \Delta(1 - \epsilon)$. Together with the result of [HMZ11], our bound of $\alpha_d$ implies that for a fixed value of $d$, we can compute $d$-sensitive optimal strategies of a deterministic game in pseudo-polynomial time, extending a theorem of [ZP96] (for the $d = -1$ case, then the bound is optimal).

**Corollary 3.4.** *If the game $\Gamma$ satisfies Assumption 1.1 and is deterministic ($M = 1$), we have $\alpha_{-1} \leq 1 - \dfrac{1}{O(Wn^3)}$ and $\alpha_{\mathsf{Bw}} \leq 1 - \dfrac{1}{24W\binom{2n}{n}}$.*

**Multiplicity approach.** In this approach, we combine our bound on $\alpha_{\mathsf{d}}$ from Theorem 3.3 with a new result on the smallest integer $d$ such that $d$-sensitive optimal strategies are also Blackwell optimal. Our next theorem parametrizes the value of such $d$ by $n$ and $W$.

**Theorem 3.5.** *Assume the game $\Gamma$ satisfies Assumption 1.1 and is deterministic. Then, there exists a constant $a > 0$ such that $\bar{d}^{\mathrm{det}}(n, W)$-sensitive optimal strategies are Blackwell optimal, where $\bar{d}^{\mathrm{det}}(n, W) := a\sqrt{(2n - 1)(1 + \log(12W))} - 2$.*

Note that $\bar{d}^{\mathrm{det}}(n, W) = O\left(\sqrt{n(1 + \log(W))}\right)$, which may be much smaller than $n - 2$ (proved by [Vei69] for MDPs) in some regimes where $\log(W) = o(n)$. To show that $d = \bar{d}^{\mathrm{det}}(n, W)$ suffices for Blackwell optimality, we show that if a pair of strategies is $\bar{d}^{\mathrm{det}}(n, W)$-sensitive optimal, it is also $d$-sensitive optimal for all $d \geq \bar{d}^{\mathrm{det}}(n, W)$, so that it is $d$-sensitive optimal for $d = -1, 0, \ldots,$

therefore it is Blackwell optimal. A key ingredient in our proof is a bound on the multiplicity of 1 as a root of a polynomial as a function of its degree and the size of its coefficients [BEK99], used to bound the multiplicity of 1 as a root of $\Delta(\alpha)$. Theorem 1.5 follows directly from the bound on $\alpha_d$ of Theorem 3.3 by choosing $d = \bar{d}^{\text{det}}(n, W)$. We refer to Table 1 for comparisons between the bounds obtained in this section, which improve by $\Omega(n)$ (for $-\log(1 - \alpha_{\text{Bw}})$) compared to previous works.

## 3.2 Separation based on Mahler measures

We now present a bound on $\alpha_{\text{Bw}}$ based on the *Mahler measure* of algebraic numbers [Leh33, Mah62].

**Definition 3.6.** The Mahler measure $M(P) \in \mathbb{R}$ of a polynomial $P$ is given by $M(P) := a \prod_{i=1}^d \max\{1, |z_j|\}$ if $P$ factorizes over the complex numbers as $P = a \prod_{i=1}^d (x - z_i)$. The Mahler measure $M(z) \in \mathbb{R}$ of an algebraic number $z$ is the Mahler measure of its minimal polynomial.

Mahler measures have applications in areas like polynomial factorization, Diophantine approximation, and knot theory, see [Smy08]. Mignotte and Waldschmidt [MW94] and Dubickas [Dub95] use them to separate algebraic numbers and 1. The following is Theorem 1 of [Dub95].

**Theorem 3.7.** *Let $\epsilon > 0$. There exists a constant $D_\epsilon \in \mathbb{N}$ such that any algebraic number $z$ of degree $d > D_\epsilon$ which is not a root of unity satisfies:*

$$|z - 1| > e^{-(\pi/4 + \epsilon)\sqrt{d \log d \log M(z)}}. \tag{4}$$

To apply Theorem 3.7, we need an estimate of $M(z)$, where $z$ is a root of $\Delta$. Since $\Delta(z) = 0$, the minimal polynomial of $z$ divides $\Delta$, and so $M(z) \le M(\Delta)$. A classical result of Landau [Lan05] states that for any complex polynomial $P = \sum_{k=0}^d c_k x^k$ we have $M(P) \le \sqrt{\sum_{k=0}^d |c_k|^2}$. Applying this to the polynomial $\Delta$ (whose coefficients and degree are bounded in Lemma 3.1), we arrive at the following upper bound for $\alpha_{\text{Bw}}$.

**Theorem 3.8.** *If the game $\Gamma$ satisfies Assumption 1.1 and is deterministic, for each $\epsilon > 0$ there exists a constant $a_\epsilon > 0$ such that the Blackwell threshold $\alpha_{\text{Bw}}$ satisfies $\alpha_{\text{Bw}} \le \alpha_{\text{Ma}}^{\text{det}}$, where*

$$-\log(1 - \alpha_{\text{Ma}}^{\text{det}}) = \max\left\{ (\frac{\pi}{4} + \epsilon)\sqrt{(2n-1)\log(2n-1)\log(12\sqrt{2}\sqrt{n}W)}, a_\epsilon + \log(12\sqrt{2}\sqrt{n}W) \right\}.$$

The maximum in Theorem 3.8 is necessary because for a fixed $\epsilon > 0$, the bound in Theorem 3.7 only applies to algebraic numbers with a degree greater than $D_\epsilon$. Thus, if a root of $\Delta$ has degree greater than $D_\epsilon$, we can apply Theorem 3.7, otherwise we apply the Lagrange bound.

**Remark 3.9.** When $W$ is "small" ($W = n^{O(1)}$), we obtain from the definition of $\alpha_{\text{Ma}}^{\text{det}}$ that $-\log(1 - \alpha_{\text{Ma}}^{\text{det}}) = O(\sqrt{n}\log n)$. When $W$ is "large" ($W = \exp(\Omega(n \log^2 n))$), we have $-\log(1 - \alpha_{\text{Ma}}^{\text{det}}) = O(\log W + \frac{\log n}{2})$, the $(\log n)/2$ term being of a lower order. There is an intermediate regime in which $n^{\Omega(1)} \le W \le \exp(O(n \log^2 n))$ and for which $-\log(1 - \alpha_{\text{Ma}}^{\text{det}}) = O(\sqrt{n \log n \log W})$.

Note that the bound on $\alpha_{\text{Bw}}$ from Theorem 3.8 is incomparable to the one of Corollary 3.4: depending on the value of $W$, none of the bounds dominates the other. Taking the best (smallest) of the two bounds, we arrive at Table 3 in which we bound the Blackwell threshold depending on the different regimes for $\log(W)$ as a function of the number of states $n$ (up to terms of lower order).

**Remark 3.10.** Using the preprocessing algorithm of Frank and Tardos [FT87], for any deterministic mean-payoff game with $n$ states of Min and $m$ states of Max, and arbitrary rational weights, we can construct a mean-payoff game with integer weights such that $W = 2^{O(nm)^3}$ and which has the same optimal strategies. For $n = m$, this estimate of $W$ is larger than the separation order $W = \exp(\Theta(n \log^2 n))$ between the "intermediate" and "high $W$" regimes in Table 3. So, all the regimes in Remark 3.9 and Table 3 are relevant.

# 4 Results for perfect-information stochastic games

We now focus on the general case of perfect-information SGs. We start by studying the structure of the discounted value function $v_i^{\sigma,\tau}(\alpha)$ associated with a pair of stationary strategies $(\sigma, \tau)$ and a state

Table 3: Bound for the Blackwell threshold $\alpha_{\mathsf{Bw}}$ of deterministic perfect-information SGs in different regimes of $\log(W)$ with respect to the number of states $n$.

| $\log W$ : | $\Theta(\log n)$ | | $\Theta(\frac{n}{\log n})$ | $\Theta(n)$ | $\Theta(n \log n)$ | | $\Theta(n \log^2 n)$ |
|---|---|---|---|---|---|---|---|
| $-\log(1-\alpha_{\mathsf{Bw}})$: | $\sqrt{n}\log n$ | $\sqrt{n\log n \log W}$ | | $n\log 2$ | $\log W$ | $\sqrt{n\log n \log W}$ | $\log W$ |

$i$. It is well-known that this function is rational, i.e., it is the ratio of two polynomials (e.g., Lemma 10.1.3 in [Put14]). We define $\Delta(\alpha)$ as the numerator appearing in $v_i^{\sigma,\tau}(\alpha) - v_i^{\sigma',\tau'}(\alpha)$ (we refer to Appendix C for the precise definition of $\Delta$). Thus, to study the zeros of $\alpha \mapsto v_i^{\sigma,\tau}(\alpha) - v_i^{\sigma',\tau'}(\alpha)$ we can focus on studying the roots of $\Delta$. We first analyze the degree and the size of the coefficients of $\Delta$.

**Proposition 4.1.** *Under Assumption 1.1, the polynomial $\Delta$ can be written as $\Delta(\alpha) = \sum_{k=0}^{2n-1} c_k \alpha^k$, where $|c_k| \leq 2nWM^{2n-1}\binom{2n-1}{k}$ for all $k \in \{0, \ldots, 2n-1\}$.*

Proposition 4.1 improves upon the corresponding results of [GCP23] for MDPs, which show that $|c_k| \leq 2nWM^{2n}4^n$ across all $k \in \{0, \ldots, 2n-1\}$.

**Results based on the Lagrange bound.** Applying the Lagrange bound to the polynomial $\epsilon \mapsto \Delta(1 - \epsilon)$, we prove the second part of Theorem 1.2. We start with the bound on $\alpha_{\mathsf{Bw}}$.

**Corollary 4.2.** *Under Assumption 1.1, we have $\alpha_{\mathsf{Bw}} \leq 1 - \dfrac{2^{\lfloor \frac{2}{3}n \rfloor - 2}}{nW(2M)^{2n-1}\binom{2n-1}{\lfloor \frac{2}{3}n \rfloor}}$ .*

We now state the bound on $\alpha_{\mathsf{d}}$ for non-deterministic SGs. Note that in the next result, we assume that the SG is unichain, i.e., that the Markov chain induced by any pair of stationary strategies is unichain. This assumption is necessary to precisely connect the coefficients of $\Delta(\alpha)$ with the coefficients of the Laurent series expansion of $v_i^{\sigma,\tau}(\alpha) - v_i^{\sigma',\tau'}(\alpha)$.

**Corollary 4.3.** *If $\Gamma$ satisfies Assumption 1.1 and is unichain, the d-sensitive threshold $\alpha_{\mathsf{d}}$ satisfies*
$$\alpha_{\mathsf{d}} \leq 1 - \frac{2^{\min\{d+2, \lfloor \frac{2}{3}n-1 \rfloor\}-1}}{nW(2M)^{2n-1}\binom{2n-1}{\min\{d+2, \lfloor \frac{2}{3}n-1 \rfloor\}+1}} .$$

**Remark 4.4** (On the unichain assumption)**.** In order to prove Corollary 4.3, the unichain assumption allows us to relate the function $\alpha \mapsto v_i^{\sigma,\tau}(\alpha) - v_i^{\sigma',\tau'}(\alpha)$ with the polynomial $\Delta(\alpha)$ using a manageable structure. More precisely, thanks to the unichain assumption, we can write

$$\Delta(\alpha) = (1-\alpha)^2 Q(\alpha)(v_i^{\sigma,\tau}(\alpha) - v_i^{\sigma',\tau'}(\alpha))$$

where $Q(\alpha)$ is a polynomial that satisfies $Q(1) \neq 0$. It follows that, for any $d'$, we have

$$\lim_{\alpha \to 1^-} (1-\alpha)^{-d'}(v_i^{\sigma,\tau}(\alpha) - v_i^{\sigma',\tau'}(\alpha)) = 0 \iff \lim_{\alpha \to 1^-} (1-\alpha)^{-(d'+2)}\Delta(\alpha) = 0 ,$$

a property that is useful to analyze the $d$-sensitive optimality (see Appendix C for more details).

Controlling this polynomial $Q(\alpha)$ in all generality (for multichain models) is the main difficulty in removing the unichain assumption. Generalizing to the multichain setting would require a finer characterization of the spectral properties of the induced Markov chains under arbitrary strategies, which is a challenging open problem.

**Results based on the Mahler bound.** Next, we present the results for $\alpha_{\mathsf{Bw}}$ based on the Mahler bound. The proof follows the same lines as in the deterministic case (Theorem 3.8).

**Theorem 4.5.** *If the game $\Gamma$ satisfies Assumption 1.1, for each $\epsilon > 0$ there exists a constant $a_\epsilon$ such that the Blackwell threshold $\alpha_{\mathsf{Bw}}$ satisfies $\alpha_{\mathsf{Bw}} \leq \alpha_{\mathsf{Ma}}$, where $-\log(1-\alpha_{\mathsf{Ma}}) = \max\{\beta_1, \beta_2\}$, $\beta_1 := (\frac{\pi}{4} + \epsilon)\sqrt{(2n-1)\log(2n-1)\log(L)}$, $\beta_2 := a_\epsilon + \log(L)$, and $L := 2nWM^{2n-1}\sqrt{\binom{2(2n-1)}{2n-1}}$.*

We end this section with a discussion on the *multiplicity approach* highlighted in Theorems 1.5 and 3.5, applied to the case of general SGs.

It is worth emphasizing that the multiplicity approach fails for general SGs. Indeed, this approach relies on bounding the multiplicity of 1 as a root of the polynomial $\Delta(\alpha)$ representing the numerator

of the functions $\alpha \mapsto v_i^{\sigma,\tau}(\alpha) - v_i^{\sigma',\tau'}(\alpha)$ (since this multiplicity leads to a bound on the minimal value of $d$ satisfying that any $d$-sensitive optimal strategy is also Blackwell optimal).

Theorem 2.1 of [BEK99] shows that given a polynomial $P = \sum_{k=0}^{d} c_k x^k$ of degree $d$ such that $\max_k |c_k| \leq 1$, the multiplicity $\bar{d}$ of 1 as a root of $P$ is bounded by $O\left(\sqrt{d(1 + \log |c_0|)}\right)$. For deterministic games, we can ensure that $|c_0|$ remains small, allowing us to obtain the meaningful bound $\bar{d} = O(\sqrt{n})$ by this technique (see Lemma 3.1 and Theorem 3.5). However, for general stochastic instances, the magnitude of $|c_0|$ can grow as $O(n4^n M^n)$. In particular, under Assumption 1.1, using arguments similar to the ones used to prove Theorem 3.5, we get that every $\bar{d}^{\mathrm{sto}}(n, W)$-sensitive optimal strategy is Blackwell optimal, where

$$\bar{d}^{\mathrm{sto}}(n, W) := a\sqrt{(2n-1)\left(1 + \log\left(2nWM^{2n-1}\binom{2n-1}{n}\right)\right)}$$

for some $a > 0$. This leads to a vacuous bound ($\bar{d} = \Omega(n)$) that is not better than the existing one. Indeed, it is known that $d = n - 2$ is enough [Vei69], and $n - 2$ is much smaller than $\bar{d}^{\mathrm{sto}}(n, W)$. This shows that while the multiplicity approach may be useful in the deterministic case (in the sense that there are some regimes of $n$ and $W$ where $\bar{d}^{\mathrm{det}}(n, W) < n - 2$), it does not yield any improvement over existing bounds in the stochastic case, highlighting the strength of our new results based on the Lagrange and Mahler bounds.

## 5    Discussion

We obtain bounds on the Blackwell threshold $\alpha_{\mathsf{Bw}}$ and on the $d$-sensitive threshold $\alpha_{\mathsf{d}}$. We improve the existing bounds on $-\log(1 - \alpha_{\mathsf{Bw}})$ by a factor $\Omega(n)$ (compared to [AM09] for SGs and [GCP23] for MDPs), and we provide the first bound on $\alpha_{\mathsf{d}}$ beyond the case $d = -1$ in deterministic games. Blackwell and mean-payoff optimal strategies have received some attention in reinforcement learning and SGs in recent years, and our bounds control the complexity of the main method to compute Blackwell and $d$-sensitive optimal strategies for SGs, by choosing $\alpha > \alpha_{\mathsf{Bw}}$ (or $\alpha > \alpha_{\mathsf{d}}$) in algorithms for solving discounted SGs. A crucial advantage of this approach is that our bounds can be combined with *any* progress in solving discounted SGs. However, to the best of our knowledge, all algorithms whose complexity depends on the discount factor $\alpha$ scale as $\tilde{O}((1 - \alpha)^k)$ for some negative $k$ (e.g. $k = -1$ for strategy iteration [Ye11, HMZ13, AG13], or $k = -3$ for sampling-based methods [SWYY20]). Our bounds on $(1 - \alpha_{\mathsf{Bw}})^{-1}$ involve some terms that grow superpolynomially in $n$ (see Tables 1 and 2). Obtaining stronger bounds for $\alpha_{\mathsf{Bw}}$ is an important future research direction.

*Open questions.* It would be interesting to obtain bounds for $\alpha_{\mathsf{d}}$ for general (non-deterministic) perfect-information SGs, *without* the unichain assumption. Additionally, our work provides several different upper bounds for $\alpha_{\mathsf{Bw}}$, all of which may be exponentially close to 1. It is essential to understand if this is a limitation of our line of analysis - for instance, does there exist stronger separation results between a root of a polynomial $P$ and 1 for the specific polynomials $\Delta$ appearing in our proof? -, or if this is an inherent difficulty of the problem. Obtaining lower bounds for $\alpha_{\mathsf{Bw}}$ and $\alpha_{\mathsf{d}}$ in all generality is an important next step.

### Acknowledgements

We thank the anonymous reviewers for their constructive comments on improving the paper. Julien Grand-Clément was supported by Hi! Paris and Agence Nationale de la Recherche (Grant 11-LABX-0047).

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

# A  Bounds from previous work

**Bounds from [AM09].**  The authors in [AM09] focus on perfect-information SGs.  Lemma 1 of [AM09] shows that $\alpha_{\mathsf{Bw}} \leq 1 - 2(n!)^2 4^n \max\{M, W\}^{2n^2}$.  We then apply the classical bound $n! = O\left(\sqrt{n}\left(\frac{n}{e}\right)^n \exp\left(\frac{1}{12n}\right)\right)$ to obtain the bound presented in Tables 1 and 2.

**Bounds from [GCP23].**  The authors of [GCP23] focus on MDPs.  Their main bound for $\alpha_{\mathsf{Bw}}$ is given in Theorem 4.4, and their bound can be simplified to $-\log(1 - \alpha_{\mathsf{Bw}}) = O\left(n\log(W) + n^2(1 + \log(M))\right)$, see the calculation in Appendix E of [GCP23].

**Bounds from [MK25].** The bound for $-\log(1 - \alpha_{\mathsf{Bw}})$ is given in Section 4.4 of [MK25], with the notation $b$ for our term $\log(W)$.

# B  Proofs for Section 3

In this section we provide the proofs of the results of Section 3. We first prove Lemma 3.1.

*Proof of Lemma 3.1.*  Note that the polynomial $\Delta$ defined in (3) satisfies

$$\Delta(\alpha) = (1 - \alpha^q)(1 - \alpha^{q'})(\langle r, \pi \rangle_\alpha - \langle r, \pi' \rangle_\alpha) + (1 - \alpha^{q'})\alpha^p \langle r, \gamma \rangle_\alpha - (1 - \alpha^q)\alpha^{p'}\langle r, \gamma' \rangle_\alpha \ .$$

Thus, if the absolute value of the instantaneous rewards are bounded by $W$, the coefficients of the polynomial of Lemma 3.1 satisfy $|a_k| \leq 12W$ for all $k$. In addition, we have

$$K = \max\{q + q' + \max\{p, p'\} - 1, q' + p + q - 1, q + p' + q' - 1\} \leq 2n - 1 \ ,$$

because we can choose $\pi$, $\gamma$, $\pi'$ and $\gamma'$ elementary (i.e., such that in the corresponding sequences of states, no state appears twice, except the initial and last state in the case of circuits), and so $p + q \leq n$, $p \leq n - 1$, $p' + q' \leq n$ and $p' \leq n - 1$. □

## B.1  Proofs for Section 3.1

We now provide the intermediate results that we need to prove Theorem 3.3. In what follows, we denote by $H(P)$ the height of the polynomial $P = \sum_{k=0}^{d} c_k x^k$ defined as $H(P) := \max_{k \in \{0,\ldots,d\}} |c_k|$.

**Lemma B.1.** *Let $P = \sum_{k=0}^{d} c_k x^k$ be a polynomial with integer coefficients and $Q(y) = \sum_{k=0}^{d} c_k' y^k$ be the polynomial which is obtained making the change of variable $x = 1 - y$ in $P(x)$. Suppose that, for some $j \in \{0, \ldots, d\}$, we have $c_j' \neq 0$ and $c_i' = 0$ for all $i < j$. Then, the polynomial $Q(y)$ has no zeros in the interval $]0, \frac{1}{2H(P)\binom{d+1}{j+2}}[$.*

*Proof.*  We first bound the magnitude of the coefficients of $Q$ in terms of $H(P)$. Since $Q(y) = P(1 - y)$, we have $c_i' = (-1)^i \sum_{k=i}^{d} c_k \binom{k}{i}$ for $i \in \{0, \ldots, d\}$. Now we use $\sum_{k=i}^{d} \binom{k}{i} = \binom{d+1}{i+1}$ to obtain that $|c_i'| \leq H(P)\binom{d+1}{i+1}$ for all $i \in \{0, \ldots, d\}$.

Note that $|c_j'| \geq 1$ since $c_j' \neq 0$ and $c_j' \in \mathbb{Z}$. Besides, note that for $m \geq 3$ and $j \leq m - 1$, we have

$$\binom{m}{i} \leq \binom{m}{j+1}^{i-j} \tag{5}$$

for all $i \in \{j + 1, \ldots, m\}$. Indeed, the cases $i = j + 1$ and $i = m$ are trivial, so we may assume $j < m - 1$ (because $j = m - 1$ implies $i = m$). Suppose that $\binom{m}{i} \leq \binom{m}{j+1}^{i-j}$ for some $i \leq m - 1$. Then,

$$\binom{m}{i+1} = \frac{m-i}{i+1}\binom{m}{i} \leq \frac{m}{2}\binom{m}{j+1}^{i-j} \leq \binom{m}{j+1}^{i+1-j} \ ,$$

because $\frac{m}{2} \leq \frac{m}{2}\frac{m-1}{j+1}\cdots\frac{m-j+1}{3}\frac{m-j}{1} = \frac{m}{j+1}\frac{m-1}{j}\cdots\frac{m-j+1}{2}\frac{m-j}{1} = \binom{m}{j+1}$.

We are now ready to apply the Lagrange bound (Theorem 3.2). Using (5) we get

$$\left(\frac{|c'_j|}{|c'_i|}\right)^{\frac{1}{i-j}} \geq \frac{|c'_j|^{\frac{1}{i-j}}}{H(P)^{\frac{1}{i-j}}\binom{d+1}{i+1}^{\frac{1}{i-j}}} \geq \frac{1}{H(P)\binom{d+1}{j+2}}$$

for any $i \in \{j+1, \ldots, d\}$. The lemma now follows from Theorem 3.2. $\qquad\square$

Note that the polynomial $\epsilon \mapsto \Delta(1-\epsilon)$ can be rewritten as

$$\Delta(1-\epsilon) = \sum_{i=0}^{K}(-1)^i \epsilon^i b_i \tag{6}$$

where $b_i = \sum_{k=i}^{K} a_k \binom{k}{i}$. Then, the next proposition is a direct consequence of Lemmas B.1 and 3.1.

**Proposition B.2.** *Suppose that $b_0 = \ldots = b_{j-1} = 0$ and $b_j \neq 0$ for some $j \geq 1$ in (6). Then, the polynomial $\epsilon \mapsto \Delta(1-\epsilon)$ has no zeros in the interval $]0, \frac{1}{24W\binom{K+1}{j+2}}[$.*

*Proof of Theorem 3.3.* Let $\alpha'$ be such that $1 - \frac{1}{24W\binom{2n}{\min\{d+4,n\}}} < \alpha' < 1$, and let $(\sigma^*, \tau^*)$ be a pair of discount optimal strategies for the discount factor $\alpha'$. To prove the theorem, it is enough to show that $(\sigma^*, \tau^*)$ is a pair of $d$-sensitive optimal strategies.

On the contrary, suppose that $(\sigma^*, \tau^*)$ is not a pair of $d$-sensitive optimal strategies. Then, either there exist a strategy $\tau$ of player Max and a state $i$ such that

$$\lim_{\alpha \to 1^-} (1-\alpha)^{-d}(v_i^{\sigma^*,\tau^*}(\alpha) - v_i^{\sigma^*,\tau}(\alpha)) < 0 , \tag{7}$$

or there exist a strategy $\sigma$ of player Min and a state $i$ such that

$$\lim_{\alpha \to 1^-} (1-\alpha)^{-d}(v_i^{\sigma^*,\tau^*}(\alpha) - v_i^{\sigma,\tau^*}(\alpha)) > 0 . \tag{8}$$

In the first place, assume that (7) holds. Let $d'$ be the smallest value satisfying

$$\lim_{\alpha \to 1^-} (1-\alpha)^{-d'}(v_i^{\sigma^*,\tau^*}(\alpha) - v_i^{\sigma^*,\tau}(\alpha)) < 0 . \tag{9}$$

By (7), it follows that $d' \leq d$ and that $\lim_{\alpha \to 1^-} (1-\alpha)^{-d''}(v_i^{\sigma^*,\tau^*}(\alpha) - v_i^{\sigma^*,\tau}(\alpha)) = 0$ for all $d'' < d'$.

If for $q \in \mathbb{N}$ we set $[q]_\alpha := 1 + \alpha + \cdots + \alpha^{q-1}$, then the polynomial $\Delta$ defined in (3) satisfies

$$\Delta(\alpha) = (1-\alpha^q)(1-\alpha^{q'})(v_i^{\sigma,\tau}(\alpha) - v_i^{\sigma',\tau'}(\alpha)) = (1-\alpha)^2[q]_\alpha[q']_\alpha(v_i^{\sigma,\tau}(\alpha) - v_i^{\sigma',\tau'}(\alpha)) . \tag{10}$$

Therefore, for all $d''$ we have

$$\lim_{\alpha \to 1^-} (1-\alpha)^{-d''}(v_i^{\sigma^*,\tau^*}(\alpha) - v_i^{\sigma^*,\tau}(\alpha)) = 0 \iff \lim_{\alpha \to 1^-} (1-\alpha)^{-(d''+2)}\Delta(\alpha) = 0 .$$

We conclude that $b_0 = \ldots = b_{d'+1} = 0$ and $b_{d'+2} \neq 0$ if we represent the polynomial $\Delta(1-\epsilon)$ as in (6). Thus, by (7) and Proposition B.2, it follows that $v_i^{\sigma^*,\tau^*}(\alpha) - v_i^{\sigma^*,\tau}(\alpha) < 0$ for $1 - \frac{1}{24W\binom{K+1}{d'+4}} < \alpha < 1$, and therefore this remains true for $1 - \frac{1}{24W\binom{2n}{\min\{d+4,n\}}} < \alpha < 1$ because $K + 1 \leq 2n$ and $d' \leq d$. Since $1 - \frac{1}{24W\binom{2n}{\min\{d+4,n\}}} < \alpha' < 1$, in particular we have $v_i^{\sigma^*,\tau^*}(\alpha') - v_i^{\sigma^*,\tau}(\alpha') < 0$, contradicting the fact that $(\sigma^*, \tau^*)$ is a pair of discount optimal strategies for $\alpha'$.

On the other hand, if (8) is satisfied, using symmetric arguments we also arrive to a contradiction. This completes the proof. $\qquad\square$

*Proof of Corollary 3.4.* The first inequality of this corollary readily follows from Theorem 3.3. The second inequality follows from Proposition B.2, the fact that $K \leq 2n - 1$ and that the function $j \mapsto \frac{1}{24W\binom{2n}{j+2}}$ is convex and achieves its minimum at $j = n - 2$. $\qquad\square$

We conclude this section with the proof of Theorem 3.5.

*Proof of Theorem 3.5.* Theorem 2.1 of [BEK99] shows that if $P = \sum_{k=0}^{d} c_k x^k$ is a non-zero polynomial such that $\max_{k \in \{0,\dots,d\}} |c_k| \leq 1$, then the multiplicity of 1 as a root of $P$ cannot exceed $a\sqrt{d(1 - \log |c_0|)}$, where $a > 0$ is an absolute constant. Applying this result to the polynomial $\frac{\Delta(\alpha)}{12W}$, we conclude that the multiplicity of 1 as a root of $\Delta(\alpha)$ is at most $a\sqrt{(2n-1)(1 + \log 12W)}$. Then, by (10) we can write $v_i^{\sigma,\tau}(\alpha) - v_i^{\sigma,\tau'}(\alpha)$ as $(1 - \alpha)^k Q(\alpha)$, where $k \leq \bar{d}^{\det}(n, W) := a\sqrt{(2n-1)(1 + \log 12W)} - 2$ and $Q(\alpha)$ is a continuous function satisfying $Q(1) \neq 0$.

If the pair of strategies $(\sigma, \tau)$ is $\bar{d}^{\det}(n, W)$-sensitive optimal, we have

$$0 \leq \lim_{\alpha \to 1^-} (1 - \alpha)^{-\bar{d}^{\det}(n,W)} \left( v_i^{\sigma,\tau}(\alpha) - v_i^{\sigma,\tau'}(\alpha) \right) = \lim_{\alpha \to 1^-} (1 - \alpha)^{k - \bar{d}^{\det}(n,W)} Q(\alpha) ,$$

and so $Q(1) > 0$. It follows that

$$\lim_{\alpha \to 1^-} (1 - \alpha)^d \left( v_i^{\sigma,\tau}(\alpha) - v_i^{\sigma,\tau'}(\alpha) \right) = \lim_{\alpha \to 1^-} (1 - \alpha)^{k - d} Q(\alpha) \geq 0$$

for any $d$. Using similar arguments it is possible to show also that

$$\lim_{\alpha \to 1^-} (1 - \alpha)^d \left( v_i^{\sigma',\tau}(\alpha) - v_i^{\sigma,\tau}(\alpha) \right) \geq 0$$

for any $d$ and any strategy $\sigma'$ of player Min. We conclude that $(\sigma, \tau)$ is $d$-sensitive optimal for any $d$, and so also Blackwell optimal. $\qquad\square$

## B.2 Proof for Section 3.2

*Proof of Theorem 3.8.* Given $\epsilon > 0$, let $D_\epsilon$ be the constant provided by Theorem 3.7. Let $z$ be any real root of the polynomial $\Delta$ different from 1, $d$ be its degree and $P = \sum_{k=0}^{d} c_k x^k$ be its minimal polynomial. Since $z$ is a root of $\Delta$, it follows that $P$ divides $\Delta$, and so we have $d \leq 2n - 1$ and $M(P) \leq M(\Delta)$ (see Section 1.3 of [CMP87]). Besides, note that $M(\Delta) \leq 12W\sqrt{2n}$ by Landau's bound [Lan05]. If $d > D_\epsilon$, then (4) holds, and so we have

$$|z - 1| > e^{-(\pi/4+\epsilon)\sqrt{d \log d \log M(P)}} \geq e^{-(\pi/4+\epsilon)\sqrt{(2n-1)\log(2n-1)\log M(\Delta)}}$$
$$\geq e^{-(\pi/4+\epsilon)\sqrt{(2n-1)\log(2n-1)\log(12W\sqrt{2n})}} .$$

Assume now that $d \leq D_\epsilon$. By Lemma B.1, it follows that $|z - 1| \geq \frac{1}{2H(P)\binom{d+1}{\lceil \frac{d}{2} \rceil}}$. Then, since $H(P) \leq 2^d M(P)$, we have

$$|z - 1| \geq \frac{1}{2^{d+1} M(P)\binom{d+1}{\lceil \frac{d}{2} \rceil}} \geq \frac{1}{2^{D_\epsilon+1} M(\Delta)\binom{D_\epsilon+1}{\lceil \frac{D_\epsilon}{2} \rceil}} \geq \frac{1}{2^{D_\epsilon+1}\binom{D_\epsilon+1}{\lceil \frac{D_\epsilon}{2} \rceil}12W\sqrt{2n}} .$$

Setting $a_\epsilon = \log\left( 2^{D_\epsilon+1}\binom{D_\epsilon+1}{\lceil \frac{D_\epsilon}{2} \rceil} \right)$, we conclude that $\Delta$ has no zeros in the interval $]\alpha_{\mathsf{Ma}}^{\det}, 1[$. $\qquad\square$

# C Proof for Section 4

## C.1 Proof of Proposition 4.1

We now detail the proof of Proposition 4.1.

We first provide the exact formula for the polynomial $\Delta(\alpha)$ considered in Section 4. Given a pair of stationary strategies $(\sigma, \tau)$, let $P^{\sigma,\tau}$ be the transition matrix and $r^{\sigma,\tau}$ be the vector of instantaneous rewards induced by $(\sigma, \tau)$, and let us define $Q^{\sigma,\tau} := MP^{\sigma,\tau}$ and $D^{\sigma,\tau}(\alpha) := \det(MI - \alpha Q^{\sigma,\tau})$. Since it is known that $(v_i^{\sigma,\tau}(\alpha))_{i \in [n]} = (I - \alpha P^{\sigma,\tau})^{-1} r^{\sigma,\tau}$, using Cramer's formula for the inverse

of a matrix and Laplace's cofactor extension, it follows that $v_i^{\sigma,\tau}(\alpha) - v_i^{\sigma',\tau'}(\alpha) = \frac{M\Delta(\alpha)}{D^{\sigma,\tau}(\alpha)D^{\sigma',\tau'}(\alpha)}$ for any state $i$, where

$$\Delta(\alpha) := D^{\sigma',\tau'}(\alpha)(\sum_{j=1}^{n} \mathrm{cof}_{ji}(MI - \alpha Q^{\sigma,\tau})r_j^{\sigma,\tau}) - D^{\sigma,\tau}(\alpha)(\sum_{j=1}^{n} \mathrm{cof}_{ji}(MI - \alpha Q^{\sigma',\tau'})r_j^{\sigma',\tau'}).$$

(11)

We now proceed to bound the degree and the coefficients of $\Delta$. In the next two propositions, we assume that $\frac{Q}{M}$ is a row-stochastic matrix, with $Q \in \mathbb{N}^{n \times n}$.

**Proposition C.1.** *The polynomial $\det(MI - \alpha Q)$ is of the form $\sum_{k=0}^{n} a_k \alpha^k$, where $|a_k| \le \binom{n}{k} M^n$.*

*Proof.* We have

$$\det(MI - \alpha Q) = \sum_{k=0}^{n} (-\alpha)^k M^{n-k} \mathrm{tr}(\mathcal{C}_k(Q)) \ ,$$

(12)

where $\mathcal{C}_k(Q)$ is the $k$-th compound matrix of $Q$. Since the entries of $\mathcal{C}_k(Q)$ are minors of $Q$ of size $k \times k$, and $\sum_{j=1}^{n} Q_{ij} = M$ for each $i = 1, \ldots, n$, we conclude that the absolute value of all the entries of $\mathcal{C}_k(Q)$ are less than or equal to $M^k$. The result now follows from (12) and the fact that $\mathcal{C}_k(Q)$ is of size $\binom{n}{k} \times \binom{n}{k}$. $\qquad\square$

It is worth noting that the result in the previous proposition is tight when $Q = MI$.

**Proposition C.2.** *For each $i, j \in \{1, \ldots, n\}$, the $(i, j)$ cofactor of the matrix $MI - \alpha Q$ is of the form $\sum_{k=0}^{n-1} a_k \alpha^k$, where $|a_k| \le \binom{n-1}{k} M^{n-1}$.*

*Proof.* The $(i, j)$ cofactor of the matrix $MI - \alpha Q$ is given by

$$(-1)^{i+j} \det(MJ - \alpha R) = (-1)^{i+j} \sum_{k=0}^{n-1} (-\alpha)^k M^{n-1-k} \mathrm{tr}((\mathrm{adj}_k(J))(\mathcal{C}_k(R))) \ ,$$

(13)

where $J$ and $R$ are the $(i, j)$ sub-matrices of $I$ and $Q$ respectively, $\mathrm{adj}_k(J)$ is the $k$-th higher adjugate matrix of $J$, and $\mathcal{C}_k(R)$ is the $k$-th compound matrix of $R$. Since $\mathrm{adj}_k(J)$ has just one non-zero entry per row, of absolute value one, as in the proof of Proposition C.1 we conclude that the absolute value of all the entries of $(\mathrm{adj}_k(J))(\mathcal{C}_k(R))$ are less than or equal to $M^k$. The result now follows from (13) and the fact that $(\mathrm{adj}_k(J))(\mathcal{C}_k(R))$ is of size $\binom{n-1}{k} \times \binom{n-1}{k}$. $\qquad\square$

By Propositions C.1 and C.2, both terms appearing in the polynomial $\Delta$ defined in (11) are of the form $\sum_{k=0}^{2n-1} b_k \alpha^k$, where

$$|b_k| \le nWM^{2n-1} \left( \sum_{s+l=k, s \le n-1, l \le n} \binom{n-1}{s}\binom{n}{l} \right)$$

$$= nWM^{2n-1} \left( \sum_{0 \le k-l \le n-1, l \le n} \binom{n-1}{k-l}\binom{n}{l} \right)$$

$$\le nWM^{2n-1} \left( \sum_{l=0}^{k} \binom{n-1}{k-l}\binom{n}{l} \right) = nWM^{2n-1} \binom{2n-1}{k} \ ,$$

by Vandermonde's Identity. Therefore, we can rewrite $\Delta$ as $\sum_{k=0}^{2n-1} c_k \alpha^k$, where $|c_k| \le 2nWM^{2n-1}\binom{2n-1}{k}$. This concludes the proof of Proposition 4.1.

## C.2 Other proofs for Section 4

To use the Lagrange bound, we consider the polynomial $\epsilon \mapsto \Delta(1 - \epsilon)$, which can be rewritten as

$$\Delta(1 - \epsilon) = \sum_{i=0}^{2n-1} (-1)^i \epsilon^i g_i \tag{14}$$

where $g_i = \sum_{k=i}^{2n-1} c_k \binom{k}{i}$. Then, using Proposition 4.1 and the identity $\sum_{k=q}^{m} \binom{m}{k}\binom{k}{q} = 2^{m-q}\binom{m}{q}$, we have

$$|g_i| \leq \sum_{k=i}^{2n-1} 2nWM^{2n-1}\binom{2n-1}{k}\binom{k}{i} = nW(2M)^{2n-1}2^{1-i}\binom{2n-1}{i} \tag{15}$$

for all $i \in \{0, \ldots, 2n-1\}$.

**Proposition C.3.** *Let $j$ be the smallest index such that $g_j \neq 0$ in (14). Then, the polynomial $\epsilon \mapsto \Delta(1-\epsilon)$ has no zeros in the interval $\left]0, \frac{2^{j-1}}{nW(2M)^{2n-1}\binom{2n-1}{j+1}}\right[$.*

*Proof.* As in the proof of Lemma B.1, we apply the Lagrange bound (Theorem 3.2). Note that $|g_j| \geq 1$ since $g_j \neq 0$ and $g_j \in \mathbb{Z}$. Then, for any $i \in \{j+1, \ldots, 2n-1\}$, we have

$$\left(\frac{|g_j|}{|g_i|}\right)^{\frac{1}{i-j}} \geq \frac{|g_j|^{\frac{1}{i-j}}}{(nW(2M)^{2n-1}2^{1-i})^{\frac{1}{i-j}}\binom{2n-1}{i}^{\frac{1}{i-j}}} \geq \frac{2^{i-1}}{nW(2M)^{2n-1}\binom{2n-1}{j+1}}$$

by (15) and (5). The proposition now follows from Theorem 3.2. $\qquad\square$

*Proof of Corollary 4.2.* Note that the function $j \mapsto \frac{2^{j-1}}{nW(2M)^{2n-1}\binom{2n-1}{j+1}}$ is convex and attains its minimum at $j = \lfloor \frac{2}{3}n - 1 \rfloor$. Then, by Proposition C.3, we conclude that no function $\alpha \mapsto v_i^{\sigma,\tau}(\alpha) - v_i^{\sigma',\tau'}(\alpha)$ has zeros in the interval $\left]1 - \frac{2^{\lfloor \frac{2}{3}n \rfloor - 2}}{nW(2M)^{2n-1}\binom{2n-1}{\lfloor \frac{2}{3}n \rfloor}}, 1\right[$. The corollary now follows from the discussion in Section 2. $\qquad\square$

*Proof of Corollary 4.3.* Let $(\sigma^*, \tau^*)$ be a pair of discount optimal strategies for the discount factor $\alpha'$, where $\alpha'$ satisfies $1 - \frac{2^{\min\{d+2, \lfloor \frac{2}{3}n-1 \rfloor\}-1}}{nW(2M)^{2n-1}\binom{2n-1}{\min\{d+2, \lfloor \frac{2}{3}n-1 \rfloor\}+1}} < \alpha' < 1$.

As in the proof for the deterministic case, in the first place assume that there exist a strategy $\tau$ and a state $i$ such that (7) is satisfied, and let $d'$ be the smallest value satisfying (9). By (7), it follows that $d' \leq d$ and that $\lim_{\alpha \to 1^-}(1-\alpha)^{-d''}(v_i^{\sigma^*,\tau^*}(\alpha) - v_i^{\sigma^*,\tau}(\alpha)) = 0$ for all $d'' < d'$.

If the game is unichain, there exist two polynomials $p(\alpha)$ and $q(\alpha)$ such that $p(1) \neq 0$, $q(1) \neq 0$ and $\Delta(\alpha) = (1-\alpha)^2 p(\alpha)q(\alpha)(v_i^{\sigma^*,\tau^*}(\alpha) - v_i^{\sigma^*,\tau}(\alpha))$, where $\Delta$ is the polynomial defined in (11) considering the pairs of strategies $(\sigma^*, \tau^*)$ and $(\sigma^*, \tau)$. Then, we have

$$\lim_{\alpha \to 1^-}(1-\alpha)^{-d''}(v_i^{\sigma^*,\tau^*}(\alpha) - v_i^{\sigma^*,\tau}(\alpha)) = 0 \iff \lim_{\alpha \to 1^-}(1-\alpha)^{-(d''+2)}\Delta(\alpha) = 0$$

for all $d''$. We conclude that $g_0 = \ldots = g_{d'+1} = 0$ and $g_{d'+2} \neq 0$ in (14). Thus, by (7) and Proposition C.3, it follows that $v_i^{\sigma^*,\tau^*}(\alpha) - v_i^{\sigma^*,\tau}(\alpha) < 0$ for $1 - \frac{2^{d'+1}}{nW(2M)^{2n-1}\binom{2n-1}{d'+3}} < \alpha < 1$. Since the function $j \mapsto \frac{2^{j-1}}{nW(2M)^{2n-1}\binom{2n-1}{j+1}}$ is convex and attains its minimum at $j = \lfloor \frac{2}{3}n - 1 \rfloor$, and $d' \leq d$, we conclude that $v_i^{\sigma^*,\tau^*}(\alpha) - v_i^{\sigma^*,\tau}(\alpha) < 0$ for $1 - \frac{2^{\min\{d+2, \lfloor \frac{2}{3}n-1 \rfloor\}-1}}{nW(2M)^{2n-1}\binom{2n-1}{\min\{d+2, \lfloor \frac{2}{3}n-1 \rfloor\}+1}} < \alpha < 1$.

Thus, in particular we have $v_i^{\sigma^*,\tau^*}(\alpha') - v_i^{\sigma^*,\tau}(\alpha') < 0$, contradicting the fact that $(\sigma^*, \tau^*)$ is a pair of discount optimal strategies for $\alpha'$.

Now, if we assume that there exist a strategy $\sigma$ and a state $i$ such that (8) is satisfied, using similar arguments as above we arrive at a contradiction.

This shows that $(\sigma^*, \tau^*)$ is a pair of $d$-sensitive discount optimal strategies. $\qquad\square$

**Remark C.4.** Setting $M = 1$ in our bounds for the stochastic case do not recover our results for the deterministic case (Section 3). This is because, in the case of deterministic transitions, we can exploit the "path then circuit structure" of the discounted value functions, as highlighted in Section 3.

## C.3 Proof of Theorem 4.5

*Proof.* The proof follows similar arguments to the ones in the proof of Theorem 3.8, so given $\epsilon > 0$, let $D_\epsilon$ be the constant provided by Theorem 3.7.

Let $z$ be any real root of the polynomial $\Delta$ different from 1, $d$ be its degree and $P = \sum_{k=0}^{d} c_k x^k$ be its minimal polynomial. Since $z$ is a root of $\Delta$, it follows that $P$ divides $\Delta$, and so we have $d \leq 2n - 1$ and $M(P) \leq M(\Delta)$. Besides, note that by Proposition 4.1 and Landau's bound [Lan05], we have $M(\Delta) \leq 2nWM^{2n-1}\sqrt{\binom{2(2n-1)}{2n-1}}$.

If $d > D_\epsilon$, then (4) holds, and so we have

$$|z - 1| > e^{-(\pi/4+\epsilon)\sqrt{d \log d \log M(P)}} \geq e^{-(\pi/4+\epsilon)\sqrt{(2n-1)\log(2n-1)\log M(\Delta)}}$$

$$\geq e^{-(\pi/4+\epsilon)\sqrt{(2n-1)\log(2n-1)\log\left(2nWM^{2n-1}\sqrt{\binom{2(2n-1)}{2n-1}}\right)}}.$$

Assume now that $d \leq D_\epsilon$. By Lemma B.1 it follows that $|z - 1| \geq \frac{1}{2H(P)\binom{d+1}{\lceil \frac{d}{2} \rceil}}$. Then, since $H(P) \leq 2^d M(P)$, we have

$$|z - 1| \geq \frac{1}{2^{d+1}M(P)\binom{d+1}{\lceil \frac{d}{2} \rceil}} \geq \frac{1}{2^{D_\epsilon+1}M(\Delta)\binom{D_\epsilon+1}{\lceil \frac{D_\epsilon}{2} \rceil}} \geq \frac{1}{2^{D_\epsilon+1}\binom{D_\epsilon+1}{\lceil \frac{D_\epsilon}{2} \rceil}2nWM^{2n-1}\sqrt{\binom{2(2n-1)}{2n-1}}}.$$

Setting $a_\epsilon = \log\left(2^{D_\epsilon+1}\binom{D_\epsilon+1}{\lceil \frac{D_\epsilon}{2} \rceil}\right)$, we conclude that $\Delta$ has no zeros in the interval $]\alpha_{\mathsf{Ma}}, 1[$. This shows that $\alpha_{\mathsf{Bw}} \leq \alpha_{\mathsf{Ma}}$. $\square$

