# OpenReview forum: "Thresholds for sensitive optimality and Blackwell optimality in stochastic games"
_NeurIPS.cc/2025/Conference — NeurIPS 2025 poster_

### Official Review · Reviewer_vJhT · 2025-06-18

**Clarity:** 3
**Significance:** 2
**Originality:** 3
**Rating:** 4
**Confidence:** 3

**Summary:**

This paper studies how to find d-sensitive and Blackwell optimal strategies in two-player zero-sum perfect-information stochastic games. It provides new bounds on the d-sensitive threshold ( $\alpha_d$ ) and the Blackwell threshold $\left(\alpha_{B w}\right)$. These thresholds are important because they determine the complexity of finding these strategies by solving standard discounted games.

The paper's main contributions are the first bounds for $\alpha_d$ when $d>-1$ and improved bounds for $\alpha_{B w}$ . The authors use three techniques from algebraic number theory to derive their results: Lagrange bounds, Mahler measures, and root multiplicity bounds. They present results for both deterministic and stochastic games, showing which technique is best for different parameter settings.

**Questions:**

### Questions
1. The multiplicity-based method fails for stochastic games (Remark 4.5). Can you provide more intuition on why the analysis doesn't carry over from the deterministic case?
2. The paper offers three different bounds. Is this necessary due to the nature of the problem, or could a unified theory combine these techniques for a single, stronger bound?
3. What are the main obstacles to extending the $\alpha_d$ bound to general multi-chain games without the unichain assumption?
4. Does your analysis offer any insight into what a "hard" game instance (i.e., one requiring a very high discount factor) might look like?

**Ethical Concerns:**

["NO or VERY MINOR ethics concerns only"]

**Final Justification:**

The rebuttal is satisfactory, and I am going to maintain the positive rating.

**Limitations:**

The paper correctly identifies its main limitations.
1. The bounds do not yet lead to polynomial-time algorithms.
2. The $\alpha_d$ bound for stochastic games is limited to the unichain case.
3. The analysis is entirely theoretical, lacking empirical validation.

**Paper Formatting Concerns:**

I did not find any major issues.

**Quality:**

3

**Strengths And Weaknesses:**

### Strengths
1. The paper provides the first explicit bounds for the d-sensitive threshold $\alpha_d$ (for $d>-1$ ), which is a new contribution to the field.
2. The use of advanced tools from algebraic number theory (Lagrange bounds, Mahler measures, multiplicity theorems) is innovative and leads to stronger results than prior work.
3. Clear Improvements: The new bounds on the Blackwell threshold $\alpha_{B w}$ improves over existing results, particularly in their dependence on the number of states, $n$.
4. The paper is well-organized. It clearly presents the problem, the methods, and the results. The summary tables are helpful for comparing the different bounds.

### Weaknesses
1. The improved bounds still lead to algorithms with superpolynomial complexity. Therefore, the work does not yet yield efficient, practical algorithms for finding Blackwell optimal strategies.
2. The bound on $\alpha_d$ in the general stochastic case requires a unichain assumption, which limits the result's applicability.
3. The work is entirely theoretical, with no experiments to indicate how tight the bounds are in practice.

---

> ### Author Rebuttal · Authors · 2025-07-30
>
> We thank you for taking the time to review the paper. We address your questions below.
>
> # Response to questions.
>
> **Q1: multiplicity-based approach.** The multiplicity approach relies on bounding the multiplicity $\bar{d}$ of $1$ as a root of the polynomial $\Delta(\alpha)$ representing the difference of value functions (since this multiplicity leads to a bound on the minimal value of $d$ satisfying that any $d$-sensitive optimal strategy is also Blackwell optimal). A known result shows that given a polynomial $P = \sum_{k=0}^{d} c_k X^k$ of degree $d$ such that $\max_{k} |c_k| \leq 1$, then this multiplicity is bounded by $O\left(\sqrt{d(1+\log(c_0))}\right)$ (this is in lines 567 - 569 of our paper).
>
> For deterministic games, we can ensure that $|c_0|$ remains small (using Lemma 3.1 in our submission), allowing us to obtain the meaningful bound $\bar{d} = O(\sqrt{n})$ by this technique. However, for general stochastic instances, the magnitude of $c_0$ can grow as $O(n 4^n M^n)$ (with $M$ = denominator in the transitions probabilities and $n$ = number of states). This leads to a vacuous bound ($\bar{d} = \Omega(n)$) that is no better than existing ones (it is known that any $n-2$-sensitive optimal strategy is also Blackwell optimal). This is why the multiplicity approach breaks down for general games.
>
> **Q2. On our three different approaches.** We found that our three proof techniques (Lagrange, Mahler, Multiplicity) each leverage different aspects of the problem, with different strengths and weaknesses:
> * The Lagrange bound can use the fact that some coefficients are zero (which is useful for the analysis of d-sensitive thresholds as in Theorem 3.3) and always holds.
> * The Mahler bound is stronger than Lagrange in some regimes (ex: $W = O(1)$) but not in others ($W = \exp(\Theta(n))$).
> * The multiplicity analysis provides new results for MDPs and deterministic games (a better bound on $d$ such that d-sensitive strategies are also Blackwell optimal), independent of the problem of bounding the Blackwell thresholds
>
> As evident from the previous items, there are several ingredients entering into the problems that we address in this paper. In particular, our bounds depend on:
> * the number of states and the “granularity” of the games (integers $W$ and $M$ in our paper)
> * the deterministic or stochastic character of the Markov chains induced by strategies
> * the multiplicity of $1$ as a root of some polynomials $\Delta(\alpha)$ associated with the games
>
> Additionally, we focus on several different objective functions (Blackwell optimality, d-sensitive optimality for $d=-1,...,n-2$). While a unifying theory is desirable, we found no single method that yields uniformly tight bounds across all these variables. Instead, our results map out which tool performs best in each regime (as highlighted for instance in lines 120 - 126 or in Table 3).
>
> **Q3. From unichain to multichain.** The unichain assumption allows us to neatly relate conditions on the value function differences $v^{\sigma*,\tau*}(\alpha) - v^{\sigma*,\tau}(\alpha)$ with some conditions on the polynomial $\Delta(\alpha)$ with a manageable structure (lines 636 - 639 on page 17, in Appendix C). More specifically, under the unichain assumption, we can write
> $$\Delta(\alpha) = R(\alpha) \cdot (v^{\sigma*,\tau*}(\alpha) - v^{\sigma*,\tau}(\alpha))$$ with $R(\alpha) = (1-\alpha)^{2} * D(\alpha)$ and $D(1) \neq 0$. From this, we can reformulate the definition of the d-sensitive optimality (Equation (2) in Definition 2.2 on page 5):
> $$\lim_{\alpha \rightarrow 1} (1-\alpha)^{d} (v^{\sigma*,\tau*}(\alpha) - v^{\sigma*,\tau}(\alpha) )= 0$$
> as some conditions on the polynomial $\Delta(\alpha)$:
> $$\lim_{\alpha \rightarrow 1} (1-\alpha)^{d-2} \Delta(\alpha)= 0.$$
> Controlling this polynomial $R(\alpha)$ in all generality (for multichain models) is the main difficulty in removing the unichain assumption. Generalizing to the multichain settings would require a finer characterization of the spectral properties of the induced Markov chains under arbitrary strategies, which is a challenging open problem. We will clarify this further and make it more explicit in our revised manuscript.
>
> **Q4. On hard game instances.** This is a fundamental question, which we list as a next step in our open questions subsection (page 9), where we discuss the importance of finding a lower bound on $\alpha_{\sf bw}$ (and therefore a matching instance). Our analysis suggests that the hardest instances should be *stochastic* and *multichain* and should involve intricate transition structures.
> More fundamentally, our techniques allow us to relate questions about stochastic games/MDPs (how close to $1$ can $\alpha_{\sf bw}$ be, for a stochastic game with $n$ states, integer rewards bounded by $W$, and transitions probabilities with common denominator $M$?) with questions from polynomials and algebraic number theory (how close to $1$ can a real root of a polynomial $P$ be, if $P$ has degree $n$ and its coefficients are bounded by $H$?). Thus, our approach provides a stronger connection between games and the rich literature on polynomials, which we plan to further exploit for demonstrating lower bounds.
>
> **On limitations.**
> We acknowledge the methodological nature of our work. However, we believe our work marks a substantial step forward in the analysis of long-term decision-making: we significantly improve the best-known bound for the Blackwell threshold by reducing the exponent by a full order of magnitude, and we provide the first explicit bounds for the d-sensitive thresholds. To achieve this, we introduce new analytical tools—most notably, the use of Mahler measure bounds, which had not been previously applied in this setting, and a multiplicity-based approach that sharpens our understanding of when d-sensitive strategies become Blackwell optimal. These techniques go beyond prior methods and offer tighter, more general bounds across a variety of regimes.
>
> More broadly, our results contribute to a fundamental and long-standing question in reinforcement learning and game theory: how farsighted must an agent be to act optimally in the long run? This question underpins many real-world applications, from planning under uncertainty to multi-agent systems, and sharp answers have been elusive. Our paper makes concrete progress in this direction. We view this as an essential step toward more principled long-horizon decision-making in learning systems. It is also worth emphasizing that methodological contributions to game theory (broadly defined) have been a substantial contingent of the papers accepted at ML conferences in recent years, even attracting awards, e.g., best-paper award at AAAI 2023 for mean-payoff robust MDPs [0] and ICML 2023 [1], or spotlights at NeurIPS 2022 [2,3]. These kinds of methodological advances form the backbone of many prominent successes that made the headlines in the newspaper recently, such as Deepmind solving Go [4] and CMU solving Poker [5], and even reinforcement learning-based fine-tuning of LLMs [6].
>
> # Conclusion
> We hope that we addressed your main comments. We will refine the paper based on your inputs and our response to your questions. We are looking forward to your new feedback and evaluation/score, and we remain at your disposal through the rebuttal process in case you have further questions.
>
> **References**
>
> [0] Wang, Y., Velasquez, A., Atia, G., Prater-Bennette, A., & Zou, S.. Robust average-reward markov decision processes. AAAI 2023.
>
> [1] Fiegel, C., Ménard, P., Kozuno, T., Munos, R., Perchet, V., & Valko, M.. Adapting to game trees in zero-sum imperfect information games. ICML 2023.
>
> [2] Cai, Y., Oikonomou, A., & Zheng, W.. Finite-time last-iterate convergence for learning in multi-player games. NeurIPS 2022.
>
> [3] Anagnostides, I., Farina, G., Kroer, C., Lee, C. W., Luo, H., & Sandholm, T. Uncoupled learning dynamics with $ o (\log t) $ swap regret in multiplayer games. NeurIPS 2022.
>
> [4] Silver, D., Huang, A., Maddison, C. J., Guez, A., Sifre, L., Van Den Driessche, G., ... & Hassabis, D.. Mastering the game of Go with deep neural networks and tree search. Nature 2016.
> [5] Brown, N., & Sandholm, T.. Superhuman AI for heads-up no-limit poker: Libratus beats top professionals. Science 2018.
>
> [6] Munos, R., Valko, M., Calandriello, D., Azar, M. G., Rowland, M., Guo, Z. D., ... & Piot, B. (2023). Nash learning from human feedback. arXiv preprint arXiv:2312.00886, 18.

---

> > ### Comment · Reviewer_vJhT · 2025-08-02
> >
> > I thank the author for the detailed response and clear answers to my questions. I am happy to maintain the positive rating.

---

### Official Review · Reviewer_Ut7d · 2025-06-30

**Clarity:** 4
**Significance:** 4
**Originality:** 3
**Rating:** 5
**Confidence:** 4

**Summary:**

The authors study two‑player zero‑sum perfect‑information stochastic games and ask how close the discount factor
$\alpha$ needs to be to $1$ before the optimal discounted strategies are already optimal for more farsighted criteria.
In particular, they consider the notion of Blackwell optimality – strategies that remain optimal for every $\alpha$ sufficiently close to $1$, and an intermediate notion called d‑sensitive optimality that interpolates between mean‑payoff (for $d=-1$) and Blackwell optimality (for $d$ approachng infinity).
The authors use heavily algebraic tool such as Lagrange separation bound and Mahler‑measure bounds to derive new thresholds on $\alpha$ that yield substantial quantitative improvements compared to the bounds in the prior works.

**Questions:**

1. Does author have some guess on whether the fact that some bounds are exponentially close to $1$ are in inherent or limitations of the analysis.
2. Adding game instances where thresholds are close to $1$ would help calibrate how tight the theory is.

**Ethical Concerns:**

["NO or VERY MINOR ethics concerns only"]

**Final Justification:**

The reviewer has adequately addressed my questions. My attitude remains positive for the paper.

**Limitations:**

Yes.

**Paper Formatting Concerns:**

No.

**Quality:**

4

**Strengths And Weaknesses:**

Strengths.
1. The paper uses sophisticated algebraic methods such as Lagrange bounds and Mahler‑measure techniques.
2. The explicit bounds improve previous results by up to a factor of $Ω(n)$, where $n$ is the size of the game, for the Blackwell optimality threshold, and, for the first time, bound the threshold for $d$-sensitive optimality for general values of $d$. This tightens the best-known estimates for the complexities of computing Blackwell/$d$-sensitive optimal strategies.
3. The paper is technical in nature: even with the connection between Blackwell threshold and classic polynomial root separation results, to apply these results, non-trivial analysis is needed to control the coefficients of the polynomial showing up in differences of discounted value functions.

Weaknesses.
1. The paper does not construct explicit games that witness a gap between their upper bounds and information‑theoretic lower bounds. It will be nice if the authors can provide even some sub-optimal game construction to show how far we are from the optimal bounds for these thresholds.
2. For stochastic game, the need for unichain structure limits applicability; relaxing this assumption is an open direction acknowledged by the authors.

---

> ### Author Rebuttal · Authors · 2025-07-30
>
> We thank you for your time reviewing our paper and for your positive evaluation.
>
> **Response to your questions.** Both of your questions pertain to the tightness of the bounds that we obtain. Our *conjecture* is that the bounds based on Mahler measures are tight, at least for deterministic games. More precisely, we believe that there exist some game instances where the Blackwell threshold $\alpha_{\sf bw}$ is exponentially close to $1$, with a gap close to the bound of Theorem 1.3, but we do not have a proof of this fact yet.
>
> This conjecture is based on the known tightness of the Mahler bound for polynomials. It is shown in Dubickas~[0] that there exist real polynomials that have *complex* roots exponentially close to $1$. However, whether this is true for *real* roots remains an open question. Indeed, in the work of Dubickas and others, the existence of complex roots close to one is obtained by a coarse control of the ``Newton polygons’’, depending on the log of moduli of the coefficients of candidate polynomials. Showing the existence of real roots close to $1$ requires a finer control (of the log of moduli of the coefficients, and also on the sign), which we do not have yet because some of the candidate polynomials are obtained in a non-explicit way (as an application of a lemma of Siegel on the existence of small integer solutions of underdetermined linear systems). Whether the bounds remain tight in the case of multichain stochastic games is also an open question. We highlight this as an important future direction of research in our open questions subsection on page 9.
>
> We do agree that exploring explicit game instances where the thresholds are near to $1$ would be valuable and give more insight on d-sensitive and Blackwell optimality. Constructing such examples—whether worst-case, drawn from real-world scenarios, or sampled at random—is an exciting avenue we plan to investigate further.
>
> [0] Dubickas, A. (1998). On algebraic numbers close to 1. Bulletin of the Australian Mathematical Society, 58(3), 423-434.
>
> **Conclusion:**
> We hope that we addressed your main comments, and we thank you again for your positive score. We will refine the paper's presentation based on your input. We remain at your disposal throughout the rebuttal process in case you have further questions.

---

> ### Comment · Reviewer_Ut7d · 2025-08-05
> **Rebuttal Response**
>
> Thank you for the response. My attitude remains positive for the paper.

---

### Official Review · Reviewer_zUfR · 2025-07-02

**Clarity:** 3
**Significance:** 3
**Originality:** 3
**Rating:** 5
**Confidence:** 2

**Summary:**

The paper obtains bounds on the d-sensitive ($\alpha_d$) and Blackwell thresholds ($\alpha_{BW}$) for stochastic games. The bounds improve upon existing results. Deterministic and general stochastic games are considered.

**Questions:**

1. The clarification regarding MDPs mentioned in the weaknesses section can be clarified.
2. Is there an assumption of smoothness on the rewards function in either the MDP or the stochastic games? Or is it not needed due to the finite state space?

**Ethical Concerns:**

["NO or VERY MINOR ethics concerns only"]

**Limitations:**

Yes.

**Paper Formatting Concerns:**

The paper is formatted well.

**Quality:**

3

**Strengths And Weaknesses:**

Strengths :

1. A thorough analysis of the different regimes of the stochastic games parameter is provided.
2. A variety of techniques are used to show the bounds.


Weaknesses:

1. The comparison tables can be improved - the results of MDP should be presented separately from those of stochastic games.
2. Since the MDPs may be such that the reward function depends on both the current and next states maybe a clarification for the definition of MDPs considered can be detailed, along with its affects on the results on $\alpha_{bw}$.

---

> ### Author Rebuttal · Authors · 2025-07-30
>
> We appreciate your time in reviewing our paper and we thank you for your positive score. We answer your questions below.
>
> **Question 1: clarification**
>
> - Thanks for pointing out these presentation issues. In our revised manuscript, we will use the extra space to enhance the exposition of past results and more clearly separate and highlight the differences in the results obtained for MDPs (one player) and stochastic games (two players).
>
> - We have introduced stochastic games where the rewards $r_{i}^{ab}$ depend on the current state $i$ and the players’ actions $a,b$, but not on the next state $j$. This assumption is standard and without loss of generality - any game with rewards of the form $r_{ij}^{ab}$ can be converted to a game with rewards dependent only on current-state-action triples $(i,a,b)$ by expanding the state space to include intermediary states of the form (i,j,a,b). It is worth emphasizing that many standard textbooks and papers on stochastic games focus on the setting with $r_{i}^{ab}$ instead of $r_{ij}^{ab}$, e.g. the competitive MDP textbook [0] (e.g. chapter 3 there on discounted stochastic games) and the main collection of articles on stochastic games collected by Neyman and Sorin [1] (e.g. Chapter 1 by Lloyd Shapley in this book).
>
>
> [0] Filar, J., & Vrieze, K. (2012). Competitive Markov decision processes.
>
> [1] Neyman, A., & Sorin, S. (Eds.). (2003). Stochastic games and applications.
>
> **Question 2: smoothness**
>
> You are correct in noting that we do not require any smoothness assumptions in our paper. Since the sets of states and actions are finite, all functions (rewards and transitions) are discrete. Therefore, continuity or differentiability assumptions are not needed. It is worth noting that the Blackwell threshold may not exist in models with continuous state sets, and this is even the case for countable state sets, as shown in Maitra’1965 “Dynamic programming for countable state systems”. We will make this point more explicit in the paper to avoid any confusion.
>
> # Conclusion:
>
> We hope that we addressed your main comments, and we thank you again for your positive score. We will refine the paper's presentation based on your input. We remain at your disposal throughout the rebuttal process in case you have further questions.

---

> > ### Comment · Reviewer_zUfR · 2025-08-04
> > **Thanks**
> >
> > Thank-you for the response. I maintain support for the paper.

---

### Official Review · Reviewer_jDrr · 2025-07-03

**Clarity:** 3
**Significance:** 3
**Originality:** 3
**Rating:** 4
**Confidence:** 2

**Summary:**

This paper studies refinements of the mean-payoff criterion in two-player zero-sum perfect-information stochastic games, focusing on Blackwell optimality and its generalization through d-sensitive optimality. The authors introduce new bounds on the d-sensitive threshold \alpha_d​ that interpolates between mean-payoff and Blackwell optimality. The paper employs the methods and toolkits from separation bounds on algebraic numbers and shows their results can improve bounds for the Blackwell threshold. These thresholds are critical for understanding the algorithmic complexity of computing optimal strategies, as solving discounted games typically requires O(1/(1−α)) iterations.

**Questions:**

- The results seem to be more technically heavy than a usual Neurips paper, would it be more suitable for a more theoretical audience to review such as COLT or ALT?
- Perhaps I am not the right reviewer for this paper, but I wonder what could be the insights and takeaways for practitioners in their design and analysis of real-world RL algorithms?

**Ethical Concerns:**

["NO or VERY MINOR ethics concerns only"]

**Final Justification:**

Overall, I recommend an acceptance of this paper given its solid technical depth. That said, I still have concerns whether this topics is interested to a general audience of NeurIPS, but I see this as a rather minor concern.

**Limitations:**

yes

**Quality:**

3

**Strengths And Weaknesses:**

Strength

- The paper did a reasonably good job introducing to me the concepts of sensitive optimality and Blackwell
Optimality. The techniques of separation bounds, algebraic roots, Mahler measures, and multiplicity theorems from algebraic number theory are completely new to me, and I am surprised at how it can be used in a learning problem. There seems to be many new ideas in the analysis of this problem (especially over another neurips paper [GCP23]), though The technical analysis is well beyond my comfort zone, so I cannot make a good judgement call.
- The problem is well-motivated to generalize this fundamental concept of Blackwell optimality to sensitive optimality. The cited statement of “one of the pressing questions in reinforcement learning” is a good showcase.
-  The paper clearly acknowledged the limitations of their results.

Weakness

- I am not sure if the problems studied by this paper fits into the theme of Neurips, or its result might be interesting to a general audience at this conference.
- A notation table can help us quickly parse through your technical results.
- Typos: the left/right square brackets throughout this paper are sometimes not matched

---

> ### Author Rebuttal · Authors · 2025-07-30
>
> We would like to thank you for taking the time to review our paper and for your constructive comments. We will revise our manuscript to fix any typos/writing issues (e.g. the brackets issues that you mentioned) and use the extra space to add a table with notations/assumptions.
>
> ## Response to your questions:
>
> **1. Fit for NeurIPS.**
>
> We believe that the topic of our work makes our paper a good fit for NeurIPS; more generally, the problem of designing methods for long-term decision-making with strategic agents has broad relevance to the machine learning community.
>
> Indeed, repeated games, MDPs, and their generalizations (like stochastic games) form the backbone of many AI systems, including prominent successes that made the headlines in the newspaper recently such as Deepmind solving Go [4] and CMU solving Poker [5], and even reinforcement learning-based fine-tuning of LLMs [6]. The specific concepts we study (Blackwell optimality and d-sensitive optimality) are directly connected with how long-term reward is modeled and optimized, a question at the heart of reinforcement learning.
>
> From an academic standpoint, this is exemplified by the number of recent papers at NeurIPS/ICML//AAAI around the following topics: Markov games, robust MDPs, repeated games in extensive-forms, and even simpler setups such as two-player zero-sum games have all attracted a lot of attention in the last years. In fact, papers focusing on these topics even received some prizes at these conferences:
>
> * a recent paper on average-reward robust MDPs (a setup equivalent to stochastic games) won a best paper award at AAAI 2023 [0];
> * two papers on multiplayer games were oral presentations at NeurIPS 2022 [1,2];
> * a paper on adaptively solving repeated games won an outstanding paper award at ICML 2023 [3].
>
> We will take care to more explicitly connect our contributions with these lines of work in the revised version.
>
> **2. On the practical takeaways.**
>
> We will do our best to revise our paper to better highlight the main takeaways for practitioners. Our contributions in this sense are twofold:
>
> - *Better modeling of long-term reward*: The different objectives considered in this paper (Blackwell optimality and d-sensitive optimality) are refinements of the usual discounted return. Our objectives allow decision-makers to better take into account future rewards and to choose strategies that are very far-sighted (taking into account the rewards obtained after a very large number of steps).
>
> - *Quantitative takeaway*: at a high level, our main takeaway is that optimizing these long-run objectives is quite difficult but can be safely reduced to discounted problems with discount factors close to 1; our bounds on the Blackwell threshold $\alpha_{\sf bw}$ and the d-sensitive threshold $\alpha_{\sf d}$ precisely quantify how close to 1 should these discount factors be.
>
> Let us finally mention some practical domains where long-term reward matters: This is important in applications with a very long-run horizon (e.g. games played over many rounds, such as Poker [5], Atari Games [7], Diplomacy [8], etc.) or applications requiring sustainable growth objectives, e.g. applications of sequential decision-making in sustainable exploitation and harvesting [9,10].
> We will use the additional space to highlight these connections.
>
>
>
> ## Conclusion:
>
> We hope that we addressed your main comments. We will improve the paper based on your inputs. We are looking forward to your new feedback and evaluation/score, and we remain at your disposal through the rebuttal process in case you have any other questions.
>
> **References**
>
> [0] Wang, Y., Velasquez, A., Atia, G., Prater-Bennette, A., & Zou, S.. Robust average-reward markov decision processes. AAAI 2023.
>
> [1] Cai, Y., Oikonomou, A., & Zheng, W.. Finite-time last-iterate convergence for learning in multi-player games. NeurIPS 2022.
>
> [2] Anagnostides, I., Farina, G., Kroer, C., Lee, C. W., Luo, H., & Sandholm, T. Uncoupled learning dynamics with $ o (\log t) $ swap regret in multiplayer games. NeurIPS 2022.
>
> [3] Fiegel, C., Ménard, P., Kozuno, T., Munos, R., Perchet, V., & Valko, M.. Adapting to game trees in zero-sum imperfect information games. ICML 2023.
>
> [4] Silver, D., Huang, A., Maddison, C. J., Guez, A., Sifre, L., Van Den Driessche, G., ... & Hassabis, D.. Mastering the game of Go with deep neural networks and tree search. Nature 2016.
>
> [5] Brown, N., & Sandholm, T.. Superhuman AI for heads-up no-limit poker: Libratus beats top professionals. Science 2018.
>
> [6] Munos, R., Valko, M., Calandriello, D., Azar, M. G., Rowland, M., Guo, Z. D., ... & Piot, B. (2023). Nash learning from human feedback. arXiv preprint arXiv:2312.00886, 18.
>
> [7] Mnih, V., Kavukcuoglu, K., Silver, D., Rusu, A. A., Veness, J., Bellemare, M. G., ... & Hassabis, D.. Human-level control through deep reinforcement learning. Nature 2015.
>
> [8] Meta Fundamental AI Research Diplomacy Team (FAIR)†, Bakhtin, A., Brown, N., Dinan, E., Farina, G., Flaherty, C., ... & Zijlstra, M.. Human-level play in the game of Diplomacy by combining language models with strategic reasoning. Science 2022.
>
> [9] Possingham, H, & Tuck, G. Application of stochastic dynamic programming to optimal fire management of a spatially structured threatened species. In Proceedings International Congress on Modelling and Simulation, MODSIM, pages 813–817, 1997.
>
> [10] Ekeland, I, Karp, L, & Sumaila, R. Equilibrium resource management with altruistic overlapping generations, Journal of Environmental Economics and Management, Volume 70, 2015, Pages 1-16

---

> > ### Comment · Reviewer_jDrr · 2025-08-03
> >
> > I thank the author for the detailed response. My concern remains but I understand these are rather minor ones. So I am happy to maintain the positive rating.

---

### Note · Authors · 2025-08-12

We thank the reviewers for their feedback on our paper. We appreciate the constructive comments and we will implement the main recommended changes, including improving motivations and the relation with the ongoing work on algorithmic game theory published at top ML conferences, adding notation tables, and clarifying assumptions. We appreciate the positive scores and we hope that our responses have helped clarify the contributions and significance of our paper. If our revisions are found helpful, we would be grateful if this could be taken into account in the final evaluations. Thank you once again for your time and consideration.

---

### Decision · Program_Chairs · 2025-09-17

**Decision:**

Accept (poster)

**Comment:**

This paper considers two-player perfect-information zero-sum games, whereby it provides improved bounds for the Blackwell threshold, while for the $d$-sensitive threshold, it is the first work to give results beyond the $d=-1$ case. The results are in part based on a variety of non-trivial algebraic techniques, and though a potential concern was raised by Reviewer jDrr about the fit for NeurIPS, I believe that these techniques (along with the results themselves) could certainly be of general interest to the reinforcement learning and algorithmic game theory communities, whose presence at NeurIPS is well established. As the reviewers are additionally in support of the paper, I recommend acceptance to the conference.